# Genomic stability of self-inactivating rabies

**Ernesto Ciabatti\*, Ana González-Rueda, Daniel de Malmazet, Hassal Lee, Fabio Morgese, Marco Tripodi**

MRC Laboratory of Molecular Biology, Cambridge, United Kingdom

**Abstract** Transsynaptic viral vectors provide means to gain genetic access to neurons based on synaptic connectivity and are essential tools for the dissection of neural circuit function. Among them, the retrograde monosynaptic ΔG-Rabies has been widely used in neuroscience research. A recently developed engineered version of the ΔG-Rabies, the non-toxic self-inactivating (SiR) virus, allows the long term genetic manipulation of neural circuits. However, the high mutational rate of the rabies virus poses a risk that mutations targeting the key genetic regulatory element in the SiR genome could emerge and revert it to a canonical ΔG-Rabies. Such revertant mutations have recently been identified in a SiR batch. To address the origin, incidence and relevance of these mutations, we investigated the genomic stability of SiR in vitro and in vivo. We found that "revertant" mutations are rare and accumulate only when SiR is extensively amplified in vitro, particularly in suboptimal production cell lines that have insufficient levels of TEV protease activity. Moreover, we confirmed that SiR-CRE, unlike canonical ΔG-Rab-CRE or revertant-SiR-CRE, is non-toxic and that revertant mutations do not emerge in vivo during long-term experiments.

## Editor's evaluation

The authors previously described a viral tool termed 'self-inactivating rabies' to trace neural circuits with minimized cell toxicity. However, this tool acquired mutations during passage which revert the tool to previous toxicity levels. This manuscript provides clarification on how to propagate and use the tool to minimize toxicity-promoting mutations.

**\*For correspondence:**
ciabatti@mrc-lmb.cam.ac.uk
(EC);
ciabatti@mrc-lmb.cam.ac.uk (EC)

## Introduction

The development of innovative technologies to record and manipulate the activity of large populations of neurons (*Jun et al., 2017*; *Lin and Schnitzer, 2016*; *Stirman et al., 2016*; *Yizhar et al., 2011*) has had a transformative impact on systems neuroscience leading to a deeper understanding of how specific networks control essential aspects of animal behaviour (*Fadok et al., 2017*; *Kohl et al., 2018*; *Stuber and Wise, 2016*). In particular, the latest generation of molecular sensors and actuators allow researchers to visualize (*Abdelfattah et al., 2019*; *Dana et al., 2019*) and perturb (*Kato et al., 2018*; *Shemesh et al., 2017*) the activity of individual neurons with unprecedented genetic, spatial, and temporal resolution. However, strategies to express these tools in any desired neuron within a neural network structure remain scarce. Viral vectors represent the primary approach to deliver genetic materials to mammalian brains, with adeno associated viruses (AAV) rapidly becoming the primary choice to target neurons based on anatomical location, genetic identity, or projection pattern (*Chan et al., 2017*; *Tenenbaum et al., 2004*; *Tervo et al., 2016*). Nonetheless, transsynaptic viruses are the only vectors that are able to label cells based on their synaptic connectivity, permitting the functional dissection of neural circuits. Among them, the retrograde monosynaptic G-deleted Rabies virus (ΔG-Rabies) is the most sensitive and efficient transsynaptic retrograde tracer, widely used to highlight

the structural organization of neural networks in mammals (*Callaway and Luo, 2015*; *Stepien et al., 2010*; *Tripodi et al., 2011*; *Wickersham et al., 2007b*). However, its toxicity has limited its use for functional experiments. Indeed, in the past few years, several strategies have been applied trying to overcome the known toxicity of rabies vectors and extending their use for long-term functional interrogation of neural circuits: the use of different viral strains (CVS-N2c) (*Reardon et al., 2016*), the conditional destabilization of viral proteins (Self-inactivating Rabies, SiR; *Ciabatti et al., 2017*) or the deletion of essential genes other than G (ΔGL-Rabies; *Chatterjee et al., 2018*).

All these approaches have advantages and disadvantages and collectively represent important improvements in the Rabies design. For example, the use of different parental strains in ΔG-Rabies vectors provide delayed mortality and improved tropism (*Reardon et al., 2016*), but do not overcome the continuous viral replication that eventually leads to toxicity. The deletion of genes other than G gave origin to effective axonal retrograde tracers (*Chatterjee et al., 2018*) but requires the expression of multiple transgenes for transsynaptic tracing experiments via other viruses or using transgenic animals, which have yet to be fully implemented and that risk recreating a fully functional ΔG-Rabies in the starter cells. The addition of regulatory elements to the rabies genome, as in the SiR design in which the rabies nucleoprotein (N) is conditionally targeted to the proteasome by a PEST sequence, has the advantage of abolishing continuous viral replication (*Ciabatti et al., 2017*). On the other hand, the known high mutation rate of RNA viruses (*Drake and Holland, 1999*; *Sanjuán et al., 2010*) poses the risk that naturally occurring mutations could emerge to selectively inactivate the added genetic sequence, hence potentially giving origin to toxic revertant mutants.

In its original design, SiR is produced from cDNA in conditions where PEST is constantly removed by the tobacco etch virus protease (TEVp) cleavage, which should prevent accumulations of PEST-targeting mutations. While it was suggested that such PEST-targeting mutations might be an unavoidable outcome of the SiR design (*Matsuyama et al., 2019*), here we show that such mutations, in fact, only accumulate when SiR is extensively amplified in cells expressing suboptimal levels of TEVp. Conversely, minimizing the number of passages in vitro and using high-TEVp expressing production cell lines prevents any appreciable accumulation of such mutations during SiR production.

The reported findings that ΔG-Rabies-CRE showed an apparently reduced cytotoxicity (*Chatterjee et al., 2018*) led to the suggestion that the CRE expression alone could dampen the toxicity of all ΔG-Rabies vectors, and hence of the SiR-CRE as well (*Matsuyama et al., 2019*). However, the survival of a fraction of ΔG-Rabies-CRE-infected neurons in CRE-reporter mice might be explained by the presence of a few naturally occurring defective viral particles that lack one or more key viral genes (*Wiktor et al., 1977*), which could effectively recapitulate the self-inactivating behaviour purposefully engineered in the SiR virus. Indeed, here we show that CRE expression alone is ineffective in dampening toxicity and that while SiR-CRE is entirely non-cytotoxic in cortical and sub-cortical regions for several months, canonical ΔG-Rabies-CRE displays a significant toxicity in vivo.

In summary, here we investigated the genomic stability of SiR and found that when produced in cells with high levels of TEVp with few rounds of amplification PEST-targeting mutations do not accumulate to appreciable levels. As expected, revertant-free SiR-CRE viruses but not Rab-CRE or PEST-mutated SiR-CRE are entirely non-toxic. Moreover, we show that PEST-targeting mutations do not accumulate at appreciable rate in vivo.

## Results

### De novo SiR productions do not accumulate revertant mutations

SiR self-inactivation depends on the proteasomal targeting of N by the c-terminal addition of a PEST sequence. The high rate of mutation in RNA viruses ($10^{-6}$ to $10^{-4}$ substitutions per nucleotide per round of copying) (*Sanjuán et al., 2010*) could lead to the emergence of mutations targeting PEST. If these mutations generate a premature stop codon just upstream of the c-terminal PEST sequence they could effectively revert the SiR to a canonical and cytotoxic ΔG-Rabies. To address the issue of whether and/or to what extent the emergence of such 'revertant' mutants occurs, we generated eight independent SiR productions from cDNA following the protocol we previously described (*Ciabatti et al., 2017*). We produced viral genomic libraries for each preparation (50 clones/batch) for Sanger sequencing using primers carrying random octamers in order to identify individual particles (*Figure 1A–B*). Out of the 8 independent preparations for a total of 400 individually analysed particles, we did not identify

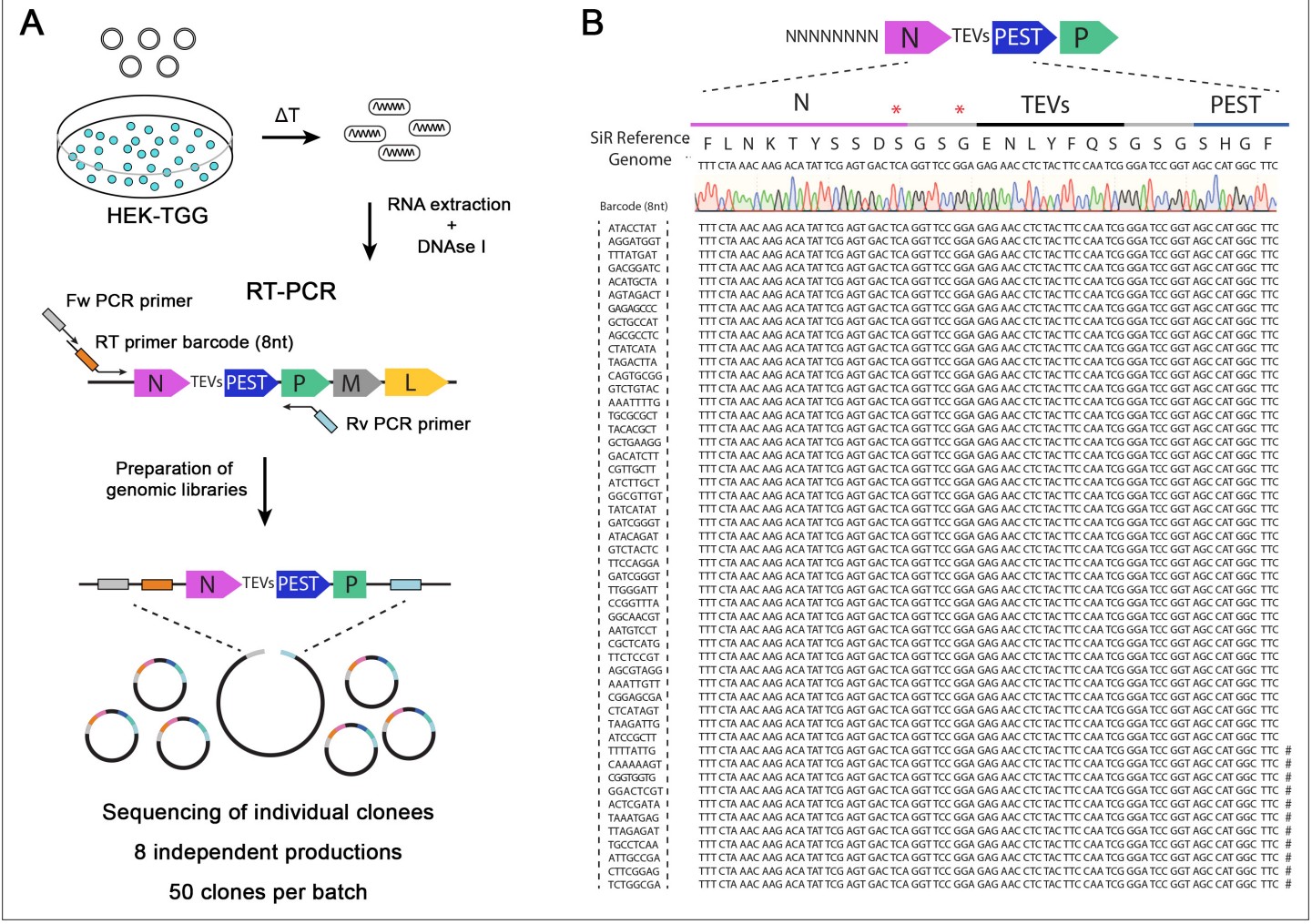

**Figure 1.** SiR production from cDNA leads to revertant-free viral preparations. (**A**) Scheme of experimental strategy to identify the emergence of "revertant" mutations during SiR production. 8 independent SiR preparations were rescued from cDNA and genomic RNA were extracted, treated with DNAse I, subjected to RT-PCR to amplify N-TEVs-PEST coding sequence and used to generate libraries for Sanger sequencing (50 clones per preparation were sequenced). (**B**) Example of sequencing results from one SiR preparation showing no mutations at the end of N. Symbols (#) show the position of previously identified mutations, marks on the sequences indicates the presence of mutations in different positions.

particles harbouring the nonsense mutations described by Matsuyama and colleagues (*Figure 1B* and *Table 1*1). The sequences' analyses showed the presence of sporadic mutations across other genomic locations (*Table 1*) as expected given the rabies mutational rate. Notably, several clones per preparation had point mutations within the N/P intergenic region, suggesting that the stoppolyadenylation signal is permissive to single base mutations (*Table 1*). These data confirm that SiRs generated from cDNA as described in *Ciabatti et al., 2017* do not accumulate mutations upstream the PEST domain at appreciable levels.

## Analysis of molecular mechanisms underpinning the potential emergence of SiR revertant mutants

Although we found no indication of emergence of PEST-targeting mutations when SiR is rescued from cDNA, a recent report finding two batches of PEST-mutated SiR (*Matsuyama et al., 2019*) unarguably points to the possibility of emergence of these mutations under certain conditions. Hence, we sought to determine which conditions might favour the accumulation of revertant mutants. In the SiR design, the PEST sequence is fused to the N protein through a cleavable linker that allows its efficient production from TEVp-expressing packaging cells (*Ciabatti et al., 2017*). The constant removal of PEST ensures that naturally occurring mutations that inactivate PEST do not provide advantage over

**Table 1.** List of detected mutations in SiR viruses rescued from cDNA divided by batch (50 individual clones per batch). The position of the mutations is calculated referring to +1 as the first base of the nucleoprotein N coding sequence.

**Sanger sequencing results of SiRs rescued from cDNA**

| | Batch A | | | | |
|---|---|---|---|---|---|
| | **Clones** | **Sequence** | **Position** | **Mutation** | **Effect on CDS** |
| Upstream N | 1/50 | GAT >GAC | –54 | Substitution | - |
| | 1/50 | AAA >AAG | –18 | Substitution | - |
| N gene | 1/50 | GCC >GCT | +186 | Substitution | Synonymous A62 |
| | 1/50 | TTT >TTTT | +243 | Insertion | Frameshift |
| | 1/50 | AAG >A-G | +485 | Deletion | Frameshift |
| | 1/50 | ATG >CTG | +562 | Substitution | Missense M188L |
| | 1/50 | GTG >G-- | +677/8 | Deletion | Frameshift |
| | 1/50 | ACG >ACCG | +983 | Insertion | Frameshift |
| | 1/50 | GAA >AAA | +1,093 | Substitution | E365K |
| | 1/50 | TCA >CCA | +1,276 | Substitution | S426P |
| TEVs-PEST | - | - | - | - | - |
| Intergenic N/P | 4/50 | AAA >AAAA | +1,571 | Insertion | - |
| | 1/50 | CCC >CCA | +1,581 | Substitution | - |
| P | - | - | - | - | - |

| | Batch B | | | | |
|---|---|---|---|---|---|
| | **Clones** | **Sequence** | **Position** | **Mutation** | **Effect on CDS** |
| Upstream N | 1/50 | AAC >A-C | –63 | Deletion | - |
| | 1/50 | CAA >CA- | –60 | Deletion | |
| | 1/50 | CTA >CTG | -3 | Substitution | - |
| N gene | 1/50 | TTT >TTTT | +243 | Insertion | Frameshift |
| | 1/50 | GAC >GAA | +501 | Substitution | D167E |
| | 1/50 | AAT >AAC | +588 | Substitution | Synonymous N196 |
| | 1/50 | GCT >GCC | +1,002 | Substitution | Synonymous A334 |
| | 1/50 | AAA >AAAA | +1,056 | Insertion | Frameshift |
| TEVs-PEST | 1/50 | TCC >TGC | +1,385 | Substitution | Missense S462C in GSG linker after TEVs |
| Intergenic N/P | 1/50 | TAT >TAA | +1,554 | Substitution | - |
| | 2/50 | AAA >AAAA | +1,571 | Insertion | - |
| P | 1/50 | GAA >GAG | +1,671 | Substitution | Synonymous E23 |
| | 1/50 | CTG >CCG | +1,775 | Substitution | Missense L58P |
| | 1/50 | GGA >TGA | +2014 | Deletion | Nonsense G138>STOP |

| | Batch C | | | | |
|---|---|---|---|---|---|
| | **Clones** | **Sequence** | **Position** | **Mutation** | **Effect on CDS** |
| Upstream N | 2/50 | AAA >AAAA | –43 | Insertion | - |
| N gene | 1/50 | TGT >TTT | +212 | Substitution | Missense C71F |
| | 1/50 | AGA >AGG | +1,074 | Substitution | Synonymous R358 |
| | 1/50 | GGT >GAT | +1,190 | Substitution | Missense G397D |

*Table 1 continued on next page*

*Table 1 continued*

| | Batch C | | | | |
|---|---|---|---|---|---|
| | **Clones** | **Sequence** | **Position** | **Mutation** | **Effect on CDS** |
| TEVs-PEST | - | - | - | - | - |
| Intergenic N/P | 1/50 | AAA >AAG | +1,569 | Substitution | - |
| | 3/50 | AAA >AAAA | +1,571 | Insertion | - |
| | 1/50 | AAA >AA- | +1,571 | Deletion | - |
| P | 1/50 | CAA >AAA | +1,720 | Substitution | Missense Q40K |
| | **Batch D** | | | | |
| | **Clones** | **Sequence** | **Position** | **Mutation** | **Effect on CDS** |
| Upstream N | - | - | - | - | - |
| N gene | 1/50 | AAG >AGG | +113 | Substitution | Missense K38R |
| | 1/50 | AAA >CAA | +295 | Substitution | Missense K99Q |
| | 1/50 | CAT >AAT | +655 | Substitution | Missense H219N |
| | 1/50 | TCA >TCC | +873 | Substitution | Synonymous S291 |
| | 1/50 | ACC >AAC | +1,196 | Substitution | Missense T399N |
| TEVs-PEST | - | - | - | - | - |
| Intergenic N/P | 3/50 | AAA >AAAA | +1,571 | Insertion | - |
| | 1/50 | ATC >ATT | +1,596 | Substitution | - |
| P | 1/50 | AAA >AAAA | +1,671 | Insertion | Frameshift |
| | 1/50 | CGT >CTA | +1,878 | Substitution | Synonymous L92 |
| | 1/50 | AGA >AGT | +1941 | Substitution | Missense R113S |
| | 1/50 | GGA >GGG | +2016 | Substitution | Synonymous G138 |
| | 1/50 | ACT >ACA | +2046 | Substitution | Synonymous T148 |
| | **Batch E** | | | | |
| | **Clones** | **Sequence** | **Position** | **Mutation** | **Effect on CDS** |
| Upstream N | 1/50 | CCA >CC- | −57 | Deletion | - |
| N gene | 1/50 | CCT >CAT | +200 | Substitution | Missense P67H |
| | 1/50 | TTT >TTTT | +243 | Insertion | Frameshift |
| | 1/50 | GGA >GAA | +371 | Substitution | Missense G124E |
| | 1/50 | ACA >ACG | +387 | Substitution | Synonymous T129 |
| | 2/50 | GAC >GAT | +393 | Substitution | Synonymous D131 |
| | 1/50 | CAC >C-- | +551/2 | Deletion | Frameshift |
| | 1/50 | ACT >AAT | +557 | Substitution | T186N |
| | 1/50 | TTT >TTTT | +779 | Insertion | Frameshift |
| TEVs-PEST | - | - | - | - | - |
| Intergenic N/P | 1/50 | CAT >CAC | +1,560 | Substitution | - |
| | 1/50 | AAA >AAC | +1,570 | Substitution | - |
| | 4/50 | AAA >AAAA | +1,571 | Insertion | - |
| | 1/50 | ATC >ATT | +1,596 | Substitution | - |
| P | 1/50 | GAA >GGA | +1,667 | Substitution | Missense E22G |

*Table 1 continued*

|  | Batch F | | | | |
| --- | --- | --- | --- | --- | --- |
|  | **Clones** | **Sequence** | **Position** | **Mutation** | **Effect on CDS** |
| Upstream N | 1/50 | ACC >AC- | –58 | Deletion | - |
|  | 1/50 | CAG >CA- | –56 | Deletion | - |
|  | 1/50 | TCA >TCG | –52 | Substitution | - |
|  | 1/50 | AAA >AAAA | –43 | Insertion | - |
|  | 1/50 | AAG >AA- | –22 | Deletion | - |
| N gene | 1/50 | TTT >TTTTT | +243/4 | Insertion | Frameshift |
|  | 1/50 | TTG >TCG | +434 | Substitution | Missense L145S |
|  | 1/50 | TTT >TT- | +534 | Deletion | Frameshift |
|  | 1/50 | GCA >GTA | +767 | Substitution | Missense A256V |
|  | 1/50 | ACA >ATA | +836 | Substitution | Missense T279I |
|  | 1/50 | AAA >AAAA | +908 | Insertion | Frameshift |
|  | 1/50 | 321 bp | +1041–1,362 | Deletion | Deletion of C-terminal of N in frame with PEST domain |
|  | 1/50 | GGA >GGG | +1,038 | Substitution | Synonymous G346 |
| TEVs-PEST | - | - | - | - | - |
| Intergenic N/P | 4/50 | AAA >AAAA | +1,571 | Insertion | - |
| P | 1/50 | CCT >CCC | +1,626 | Substitution | Synonymous P8 |
|  | 1/50 | GAA >GGA | +1,727 | Substitution | Missense E42G |
|  | 1/50 | TTT >TTC | +1,845 | Substitution | Synonymous F81 |
|  | Batch G | | | | |
|  | **Clones** | **Sequence** | **Position** | **Mutation** | **Effect on CDS** |
| Upstream N | 1/50 | CCA >CC- | –57 | Deletion | - |
|  | 1/50 | AAA >AA- | –16 | Deletion | - |
| N gene | 1/50 | GCA >GTA | +290 | Substitution | Missense A97V |
|  | 1/50 | CAT >GAT | +409 | Substitution | Missense H137D |
|  | 1/50 | TTT >TT- | +534 | Deletion | Frameshift |
|  | 1/50 | TAT >TGT | +1,271 | Substitution | Missense Y424C |
|  | 1/50 | GCC >GTC | +1,316 | Substitution | Missense A439V |
| TEVs-PEST | - | - | - | - | - |
| Intergenic N/P | 4/50 | AAA >AAAA | +1,571 | Insertion | - |
| P | 1/50 | AAA >CAA | +1,786 | Substitution | Missense K62Q |
|  | 1/50 | GAA >GGA | +1,823 | Substitution | Missense E74G |
|  | 1/50 | CGA >CAA | +1,834 | Substitution | Missense R78Q |
|  | Batch H | | | | |
|  | **Clones** | **Sequence** | **Position** | **Mutation** | **Effect on CDS** |
| Upstream N | 1/50 | AAA >AAAA | –43 | Insertion | - |
|  | 1/50 | AAC >AA- | –42 | Deletion |  |
| N gene | 1/50 | TTA >CTA | +145 | Substitution | Synonymous L49 |
|  | 1/50 | ATG >ATA | +234 | Substitution | Missense M78I |

*Table 1 continued on next page*

*Table 1 continued*

| | Batch H | | | | |
| | Clones | Sequence | Position | Mutation | Effect on CDS |
|---|---|---|---|---|---|
| | 1/50 | TTT >TTTT | +243 | Insertion | Frameshift |
| | 1/50 | AAA >CAA | +295 | Substitution | Missense K99Q |
| | 1/50 | GAT >AAT | +301 | Substitution | Missense D101N |
| | 1/50 | GGA >AGA | +622 | Substitution | Missense G208R |
| | 1/50 | GCT >TCT | +838 | Substitution | Missense A280S |
| | 1/50 | GGC >G-C | +1,028 | Deletion | Frameshift |
| | 1/50 | GAC >AAC | +1,132 | Substitution | Missense D378N |
| TEVs-PEST | 1/50 | CTG >CTA | +1,437 | Substitution | Synonymous L16 in PEST domain |
| Intergenic N/P | 3/50 | AAA >AAAA | +1,571 | Insertion | - |
| | 1/50 | AAC >AAA | +1,592 | Substitution | - |
| P | 1/50 | AAA >AAAA | +1,788 | Insertion | Frameshift |

non-mutated particles. However, we reasoned that with suboptimal TEVp activity PEST-mutants may display faster replication kinetics than SiR particles, and might eventually accumulate in the population, as in a directed-evolution experiment. Thus, we hypothesised that two factors might prominently affect the emergence of revertants: 1. low TEVp levels in packaging cells and 2. excessive rounds of amplification of SiR in vitro. First, we investigated TEVp activity in packaging cells over time by producing HEK293T cells expressing TEVp and Gsad (HEK-TGG) as previously described (*Ciabatti et al., 2017*). After selecting for TEVp-expressing cells with puromycin HEK-TGG where cultured for multiple passages in medium containing different level of antibiotic (puromycin 0 µg/ml, 1 µg/ml, 2 µg/ml; *Figure 2A*). TEVp activity was then assessed every 2 passages by transfecting a TEVp reporter (*Gray et al., 2010*) and analysing TEVp site (TEVs) cleavage by western blot (*Figure 2B*, *Figure 2—figure supplement 1*). We found that the TEVp-dependent cleavage of the overexpressed reporter decreased in HEK-TGG after amplification and by passage 6 (P6) was less than half the initial level (from 31.7±2.4% at P0 to 14.7 ± 1.7% and 13.8 ± 1.2% with 1 µg/µl and 2 µg/µl puromycin, respectively; *Figure 2B–C*). Importantly, amplification in the absence of antibiotic pressure quickly reduced TEVp activity, decreasing by one order of magnitude by P6 (31.7 ± 2.4% at P0; 7.7±1.3% at P2; 3.1±0.2% at P6 without puromycin; *Figure 2B–C*). This suggests that extensive amplification of HEK-TGG leads to selection of clones with suboptimal TEVp expression, particularly in absence of antibiotic pressure.

To test the dependence of the emergence of revertant mutations on TEVp activity in the packaging cells, and investigate the accumulation kinetics of potential mutants, we amplified four independent (sequenced) revertant-free SiR preparations in vitro in low- and high-TEVp conditions for several passages. Every two passages, genomic libraries for each viral preparation were produced by reverse-transcription of the RNA genomes using primers barcoded with unique molecular identifiers (UMI, random decamer) and PCR amplifying an amplicon containing the N-TEVs-PEST gene. Then, SiR libraries were analysed by long-read next generation sequencing (NGS) using single molecule, real-time (SMRT) PacBio technology (*Rhoads and Au, 2015*; *Figure 2D* and *Figure 2—figure supplement 1*). SMRT sequencing employs the generation of circular molecules from the N-TEVs-PEST amplicons that are replicated for several passages by a polymerase so that individual sub-reads can be combined to generate high-quality consensus sequences (sequencing accuracy ≥98% with 3 passages; *Figure 2—figure supplement 2*). Since SMRT technology is particularly prone to false-positive insertion and deletions (INDELs; *Carneiro et al., 2012*; *Dohm et al., 2020*) and all previously reported PEST-targeting mutations were substitutions (*Matsuyama et al., 2019*), we restricted our analysis to substitutions (single-nucleotide polymorphism, SNP) above 2% threshold. We considered a PEST-targeting mutation to be any non-synonymous substitution targeting either N or TEVs-PEST sequences. In accordance with our hypothesis, the extensive amplification of SiR in vitro led to

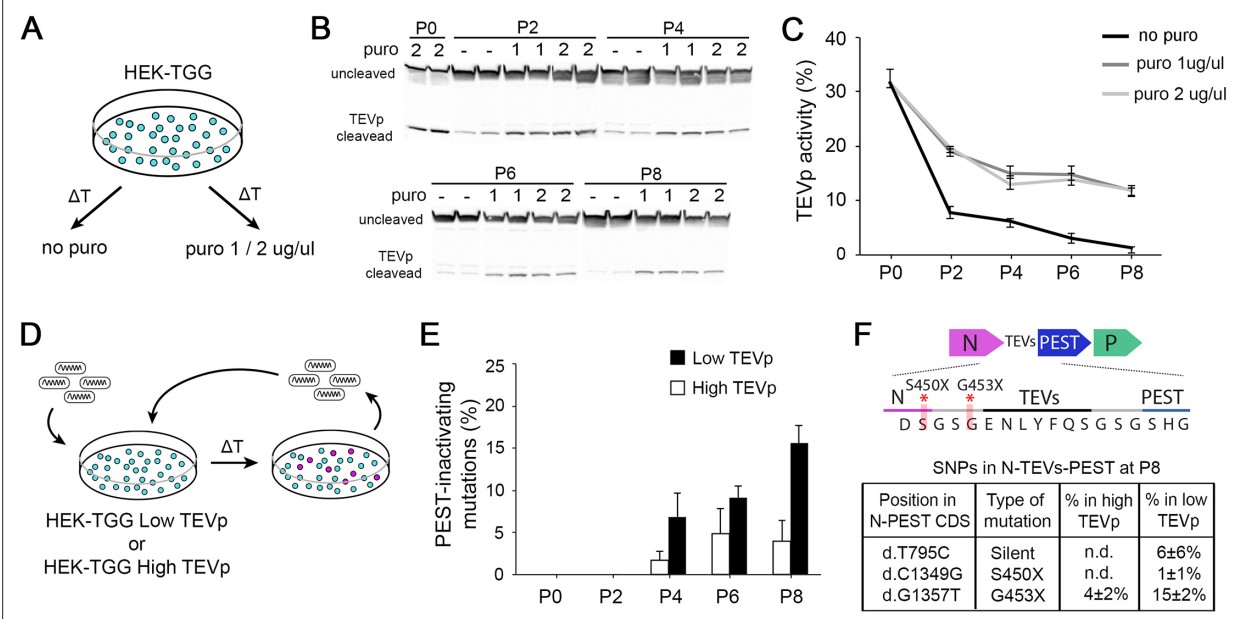

**Figure 2.** High TEVp activity in packaging cells prevents accumulation of PEST-mutations. (**A**) HEK-TGG packaging cells were amplified for several passages in absence or presence (1 or 2 µg/ml) of puromycin selection. (**B**) TEVp-dependent cleavage of TEVp-activity reporter was analysed by western blot in HEK-TGG at different amplification passages. (**C**) Quantification of TEVp-activity in packaging cells over time in presence or absence of antibiotic pressure. (mean ± SEM, n=3) (**D**) Experimental design to assess emergence of mutations in SiR preparations after multiple passages of amplification in high TEVp (HEK-TGG P0) or low TEVp HEK-TGG (HEK-TGG P8, without puromycin selection). (**E**) Quantification of frequency of the accumulation of PEST-targeting mutations over time that prevent translation of PEST domain (mean ± SEM, n=4 independent viral preparation). (**F**) Summary of the single nucleotide polymorphisms (SNPs) in the coding sequence (CDS) of N-TEVsPEST that reached threshold at P8 (mean ± SEM, n=4; n.d. indicates that the mutations were not detected above threshold). Top scheme shows the position of PEST-inactivating mutations.

The online version of this article includes the following source data and figure supplement(s) for figure 2:

**Source data 1.** Individual Western Blots used in *Figure 2B*.

**Source data 2.** TEVp-activity in HEK-TGG packaging cells over time.

**Figure supplement 1.** Western blots to test TEVp in packaging cells over time.

**Figure supplement 2.** SMRT sequencing of SiR genomic libraries.

the emergence of revertants that can accumulate within the SiR population, especially in lowTEVp packaging cells (16% ± 2% of sequences containing a revertant mutation at P8 in lowTEVp cells; *Figure 2E*, *Table 2*). On the other hand, PEST-targeting mutations remained below 5% even after 8 rounds of amplification when SiR was amplified in high-TEVp cells (4% ± 2% of sequences containing a revertant mutation at P8 in high-TEVp cells; *Figure 2E*, *Table 2*). Notably, all PEST-inactivating mutations detected in this experiment were single base substitutions introducing a premature stop codon prior to TEVs either at the last amino acid of N or immediately after (d.C1349G and d.G1357T, leading to stop insertion at S450 and G453, respectively; *Figure 2F*, *Table 2*), which also accounted for the large majority of revertant particles reported by *Matsuyama et al., 2019*. Thus, in order to avoid the accumulation of revertant mutants, SiR viruses should be only amplified in high-TEVp, low-passage packaging cells for the minimum required number of passages.

## Difference in cytotoxicity between ΔG-Rabies, PEST-mutant SiR and SiR

In the recent report of *Matsuyama et al., 2019* the authors showed that PEST-mutant SiR is cytotoxic in vivo, which is the obvious consequence of the presence of a stop codon upstream PEST that transforms the SiR into a WT ΔG-Rabies. This is strikingly different to our results showing that SiR can permanently label neurons by recombinase-mediated activation of genetic cassettes before disappearing from the infected neurons without cytotoxicity (*Ciabatti et al., 2017*). To experimentally confirm that revertant-free and PEST-mutant SiR are different viruses we characterized them in vitro and in vivo and compared them to canonical ΔG-Rabies. In order to obtain a pure preparation of PEST-mutants

**Table 2.** List of detected mutations above 2% thresholds in SiR viruses amplified in high- and low-TEVp packaging cells sequenced by SMRT NGS sequencing.

The position of the mutations is defined considering +1 the first base of the nucleoprotein N coding sequence.

**NGS sequencing results of SiRs amplified for multiple passages in vitro**

**SIR-A-P0 bc1—bc2**

|  | Position | Variant | N (q>20) | Freq % | Mutation | Effect on CDS |
|---|---|---|---|---|---|---|
| Upstream N | –49 | +A | 302/6608 | 4.5% | Insertion | - |
| N gene | +237 | +T | 266/6598 | 4.0% | Insertion | Frameshift |
|  | +636 | +T | 190/6595 | 2.9% | Insertion | Frameshift |
| TEVs-PEST | - | - | - | - | - | - |
| Intergenic | +1,564 | +A | 732/6556 | 11.1% | Insertion | - |
| P | - | - | - | - | - | - |

**SIR-B-P0 bc1—bc3**

|  | Position | Variant | N (q>20) | Freq % | Mutation | Effect on CDS |
|---|---|---|---|---|---|---|
| Upstream N | –49 | +A | 276/6045 | 4.6% | Insertion | - |
| N gene | +237 | +T | 274/6037 | 4.5% | Insertion | Frameshift |
|  | +636 | +T | 180/6036 | 3.0% | Insertion | Frameshift |
| TEVs-PEST | +1,359 | A>T | 246/5879 | 4.2% | Substitution | Silent G453 |
| Intergenic | +1,564 | +A | 729/6556 | 12.1% | Insertion | - |
| P | - | - | - | - | - | - |

**SIR-C-P0 bc1—bc4**

|  | Position | Variant | N (q>20) | Freq % | Mutation | Effect on CDS |
|---|---|---|---|---|---|---|
| Upstream N | –49 | +A | 256/5137 | 5.0% | Insertion | - |
| N gene | +237 | +T | 227/5137 | 4.4% | Insertion | Frameshift |
|  | +636 | +T | 167/5138 | 3.3% | Insertion | Frameshift |
| TEVs-PEST | - | - | - | - | - | - |
| Intergenic | +1,564 | +A | 598/5140 | 11.6% | Insertion | - |
| P | - | - | - | - | - | - |

**SIR-D-P0 bc1—bc5**

|  | Position | Variant | N (q>20) | Freq % | Mutation | Effect on CDS |
|---|---|---|---|---|---|---|
| Upstream N | –49 | +A | 249/5419 | 4.6% | Insertion | - |
| N gene | +237 | +T | 229/5419 | 4.2% | Insertion | Frameshift |
|  | +636 | +T | 125/5422 | 2.3% | Insertion | Frameshift |
| TEVs-PEST | - | - | - | - | - | - |
| Intergenic | +1,564 | +A | 612/5420 | 11.3% | Insertion | - |
| P | - | - | - | - | - | - |

**SIR-A-HighTEVp-P2 bc2—bc4**

|  | Position | Variant | N (q>20) | Freq % | Mutation | Effect on CDS |
|---|---|---|---|---|---|---|
| Upstream N | –49 | +A | 245/5934 | 4.1% | Insertion | - |
| N gene | +237 | +T | 297/5933 | 5.0% | Insertion | Frameshift |
|  | +636 | +T | 157/5938 | 2.6% | Insertion | Frameshift |
| TEVs-PEST | - | - | - | - | - | - |
| Intergenic | +1,564 | +A | 634/5935 | 10.7% | Insertion | - |

*Table 2 continued on next page*

*Table 2 continued*

**SIR-A-HighTEVp-P2 bc2—bc4**

|  | Position | Variant | N (q>20) | Freq % | Mutation | Effect on CDS |
|---|---|---|---|---|---|---|
| P | - | - | - | - | - | - |

**SIR-B-HighTEVp-P2 bc2—bc5**

|  | Position | Variant | N (q>20) | Freq % | Mutation | Effect on CDS |
|---|---|---|---|---|---|---|
| Upstream N | –49 | +A | 281/5750 | 4.9% | Insertion | - |
| N gene | +237 | +T | 272/5752 | 4.7% | Insertion | Frameshift |
|  | +636 | +T | 170/5752 | 3.0% | Insertion | Frameshift |
| TEVs-PEST | - | - | - | - | - | - |
| Intergenic | +1,564 | +A | 625/5749 | 10.9% | Insertion | - |
| P | - | - | - | - | - | - |

**SIR-C-HighTEVp-P2 bc2—bc6**

|  | Position | Variant | N (q>20) | Freq % | Mutation | Effect on CDS |
|---|---|---|---|---|---|---|
| Upstream N | –49 | +A | 236/4773 | 4.9% | Insertion | - |
| N gene | +237 | +T | 241/4772 | 5.1% | Insertion | Frameshift |
|  | +636 | +T | 137/4774 | 2.9% | Insertion | Frameshift |
| TEVs-PEST | - | - | - | - | - | - |
| Intergenic | +1,564 | +A | 489/4776 | 10.2% | Insertion | - |
| P | - | - | - | - | - | - |

**SIR-D-HighTEVp-P2 bc2—bc6**

|  | Position | Variant | N (q>20) | Freq % | Mutation | Effect on CDS |
|---|---|---|---|---|---|---|
| Upstream N | –49 | +A | 260/5591 | 4.7% | Insertion | - |
| N gene | +237 | +T | 238/5595 | 4.3% | Insertion | Frameshift |
|  | +636 | +T | 150/5597 | 2.7% | Insertion | Frameshift |
| TEVs-PEST | - | - | - | - | - | - |
| Intergenic | +1,564 | +A | 550/5594 | 9.8% | Insertion | - |
| P | - | - | - | - | - | - |

**SIR-A-LowTEVp-P2 bc1—bc6**

|  | Position | Variant | N (q>20) | Freq % | Mutation | Effect on CDS |
|---|---|---|---|---|---|---|
| Upstream N | –49 | +A | 197/3891 | 5.1% | Insertion | - |
| N gene | +237 | +T | 194/3891 | 5.0% | Insertion | Frameshift |
|  | +636 | +T | 116/3892 | 3.0% | Insertion | Frameshift |
| TEVs-PEST | - | - | - | - | - | - |
| Intergenic | +1,564 | +A | 447/3891 | 11.5% | Insertion | - |
| P | - | - | - | - | - | - |

**SIR-B-LowTEVp-P2 bc1—bc7**

|  | Position | Variant | N (q>20) | Freq % | Mutation | Effect on CDS |
|---|---|---|---|---|---|---|
| Upstream N | –49 | +A | 244/5050 | 4.8% | Insertion | - |
| N gene | +237 | +T | 227/5055 | 4.5% | Insertion | Frameshift |
|  | +636 | +T | 162/5055 | 3.2% | Insertion | Frameshift |
| TEVs-PEST | - | - | - | - | - | - |
| Intergenic | +1,564 | +A | 503/5055 | 10.0% | Insertion | - |

*Table 2 continued on next page*

*Table 2 continued*

**SIR-B-LowTEVp-P2 bc1—bc7**

|  | Position | Variant | N (q>20) | Freq % | Mutation | Effect on CDS |
|---|---|---|---|---|---|---|
| P | - | - | - | - | - | - |

**SIR-C-LowTEVp-P2 bc1—bc8**

|  | Position | Variant | N (q>20) | Freq % | Mutation | Effect on CDS |
|---|---|---|---|---|---|---|
| Upstream N | –49 | +A | 266/5050 | 5.3% | Insertion | - |
| N gene | +237 | +T | 248/5050 | 4.9% | Insertion | Frameshift |
|  | +636 | +T | 146/5056 | 2.9% | Insertion | Frameshift |
| TEVs-PEST | - | - | - | - | - | - |
| Intergenic | +1,564 | +A | 547/5054 | 10.8% | Insertion | - |
| P | - | - | - | - | - | - |

**SIR-D-LowTEVp-P2 bc1—bc9**

|  | Position | Variant | N (q>20) | Freq % | Mutation | Effect on CDS |
|---|---|---|---|---|---|---|
| Upstream N | –49 | +A | 200/5295 | 3.8% | Insertion | - |
| N gene | +237 | +T | 204/5295 | 3.9% | Insertion | Frameshift |
|  | +636 | +T | 141/5297 | 2.7% | Insertion | Frameshift |
| TEVs-PEST | - | - | - | - | - | - |
| Intergenic | +1,564 | +A | 456/5297 | 8.6% | Insertion | - |
| P | - | - | - | - | - | - |

**SIR-A-HighTEVp-P4 bc2—bc8**

|  | Position | Variant | N (q>20) | Freq % | Mutation | Effect on CDS |
|---|---|---|---|---|---|---|
| Upstream N | –49 | +A | 225/5803 | 3.9% | Insertion | - |
| N gene | +108 | +A | 154/5805 | 2.7% | Insertion | Frameshift |
|  | +237 | +T | 276/5806 | 4.8% | Insertion | Frameshift |
|  | +636 | +T | 158/5807 | 2.7% | Insertion | Frameshift |
| TEVs-PEST | +1,357 | G>T | 134/5745 | 2.3% | Substitution | Missense G453X |
| Intergenic | +1,564 | +A | 536/5803 | 9.2% | Insertion | - |
| P | - | - | - | - | - | - |

**SIR-B-HighTEVp-P4 bc2—bc10**

|  | Position | Variant | N (q>20) | Freq % | Mutation | Effect on CDS |
|---|---|---|---|---|---|---|
| Upstream N | –49 | +A | 270/5572 | 4.8% | Insertion | - |
| N gene | +237 | +T | 223/5572 | 4.0% | Insertion | Frameshift |
|  | +636 | +T | 155/5571 | 2.8% | Insertion | Frameshift |
| TEVs-PEST | - | - | - | - | - | - |
| Intergenic | +1,564 | +A | 590/5576 | 10.6% | Insertion | - |
| P | - | - | - | - | - | - |

**SIR-C-HighTEVp-P4 bc2—bc11**

|  | Position | Variant | N (q>20) | Freq % | Mutation | Effect on CDS |
|---|---|---|---|---|---|---|
| Upstream N | –49 | +A | 233/5581 | 4.2% | Insertion | - |
|  | –21 | -N | 114/5581 | 2.0% | Deletion | - |
|  | –19 | A>G | 272/5499 | 4.9% | Substitution | - |
| N gene | +237 | +T | 252/5582 | 4.5% | Insertion | Frameshift |

*Table 2 continued on next page*

*Table 2 continued*

**SIR-C-HighTEVp-P4 bc2—bc11**

|  | Position | Variant | N (q>20) | Freq % | Mutation | Effect on CDS |
|---|---|---|---|---|---|---|
|  | +636 | +T | 149/5581 | 2.7% | Insertion | Frameshift |
| TEVs-PEST | +1,357 | G>T | 248/5528 | 4.5% | Substitution | Missense G453X |
| Intergenic | +1,564 | +A | 573/5579 | 10.3% | Insertion | - |
| P | - | - | - | - | - | - |

**SIR-D-HighTEVp-P4 bc2—bc12**

|  | Position | Variant | N (q>20) | Freq % | Mutation | Effect on CDS |
|---|---|---|---|---|---|---|
| Upstream N | –49 | +A | 200/6116 | 3.3% | Insertion | - |
| N gene | +237 | +T | 219/6117 | 3.6% | Insertion | Frameshift |
|  | +636 | +T | 160/6119 | 2.6% | Insertion | Frameshift |
| TEVs-PEST | - | - | - | - | - | - |
| Intergenic | +1,564 | +A | 456/6120 | 7.5% | Insertion | - |
| P | - | - | - | - | - | - |

**SIR-A-LowTEVp-P4 bc1—bc10**

|  | Position | Variant | N (q>20) | Freq % | Mutation | Effect on CDS |
|---|---|---|---|---|---|---|
| Upstream N | –49 | +A | 239/4681 | 5.1% | Insertion | - |
| N gene | +108 | +A | 114/4682 | 2.4% | Insertion | Frameshift |
|  | +237 | +T | 242/4683 | 5.2% | Insertion | Frameshift |
|  | +636 | +T | 131/4684 | 2.8% | Insertion | Frameshift |
|  | +1,053 | +A | 97/4683 | 2.1% | Insertion | Frameshift |
| TEVs-PEST | +1,357 | G>T | 170/4650 | 3.7% | Substitution | Missense G453X |
| Intergenic | +1,564 | +A | 570/4683 | 12.2% | Insertion | - |
| P | - | - | - | - | - | - |

**SIR-B-LowTEVp-P4 bc1—bc11**

|  | Position | Variant | N (q>20) | Freq % | Mutation | Effect on CDS |
|---|---|---|---|---|---|---|
| Upstream N | –49 | +A | 255/4757 | 5.4% | Insertion | - |
| N gene | +237 | +T | 245/4758 | 5.1% | Insertion | Frameshift |
|  | +636 | +T | 141/4758 | 3.0% | Insertion | Frameshift |
| TEVs-PEST | - | - | - | - | - | - |
| Intergenic | +1,564 | +A | 551/4757 | 11.6% | Insertion | - |
| P | - | - | - | - | - | - |

**SIR-C-LowTEVp-P4 bc1—bc12**

|  | Position | Variant | N (q>20) | Freq % | Mutation | Effect on CDS |
|---|---|---|---|---|---|---|
| Upstream N | –49 | +A | 268/5461 | 4.9% | Insertion | - |
|  | –19 | A>G | 160/5403 | 3.0% | Substitution | - |
| N gene | +237 | +T | 231/5463 | 4.2% | Insertion | Frameshift |
|  | +636 | +T | 156/5466 | 2.9% | Insertion | Frameshift |
| TEVs-PEST | +1,357 | G>T | 705/5286 | 13.3% | Substitution | Missense G453X |
| Intergenic | +1,564 | +A | 538/5464 | 9.8% | Insertion | - |
| P | - | - | - | - | - | - |

*Table 2 continued on next page*

*Table 2 continued*

**SIR-D-LowTEVp-P4 bc2—bc3**

|  | Position | Variant | N (q>20) | Freq % | Mutation | Effect on CDS |
|---|---|---|---|---|---|---|
| Upstream N | –49 | +A | 266/5841 | 4.6% | Insertion | - |
| N gene | +237 | +T | 246/5838 | 4.2% | Insertion | Frameshift |
|  | +574 | -N | 140/5834 | 2.4% | Deletion | Frameshift |
|  | +636 | +T | 156/5833 | 2.7% | Insertion | Frameshift |
| TEVs-PEST | +1,357 | G>T | 200/5737 | 3.5% | Substitution | Missense G453X |
| Intergenic | +1,564 | +A | 529/5818 | 9.1% | Insertion | - |
| P | - | - | - | - | - | - |

**SIR-A-HighTEVp-P6 bc5—bc6**

|  | Position | Variant | N (q>20) | Freq % | Mutation | Effect on CDS |
|---|---|---|---|---|---|---|
| Upstream N | –49 | +A | 604/6567 | 9.2% | Insertion | - |
|  | –19 | A>G | 555/6349 | 8.7% | Substitution | - |
| N gene | +108 | +A | 227/6565 | 3.5% | Insertion | Frameshift |
|  | +166 | +T | 157/6565 | 2.4% | Insertion | Frameshift |
|  | +237 | +T | 543/6565 | 8.3% | Insertion | Frameshift |
|  | +245 | +G | 132/6565 | 2.0% | Insertion | Frameshift |
|  | +466 | +A | 175/6566 | 2.7% | Insertion | Frameshift |
|  | +636 | +T | 337/6569 | 5.1% | Insertion | Frameshift |
| TEVs-PEST | +1,357 | G>T | 767/6317 | 12.1% | Substitution | Missense G453X |
| Intergenic | +1,564 | +A | 1032/6583 | 15.7% | Insertion | - |
| P | +1,669 | +A | 155/6584 | 2.4% | Insertion | Frameshift |

**SIR-B-HighTEVp-P6 bc5—bc7**

|  | Position | Variant | N (q>20) | Freq % | Mutation | Effect on CDS |
|---|---|---|---|---|---|---|
| Upstream N | –49 | +A | 624/6752 | 9.2% | Insertion | - |
|  | –21 | -N | 202/6754 | 3.0% | Deletion | - |
|  | –20 | +G | 243/6754 | 3.6% | Insertion | - |
|  | –19 | A>G | 1180/6296 | 18.7% | Substitution | - |
| N gene | +108 | +A | 216/6752 | 3.2% | Insertion | Frameshift |
|  | +166 | +T | 185/6751 | 2.7% | Insertion | Frameshift |
|  | +237 | +T | 559/6751 | 8.3% | Insertion | Frameshift |
|  | +245 | +G | 138/6751 | 2.0% | Insertion | Frameshift |
|  | +466 | +A | 197/6753 | 2.9% | Insertion | Frameshift |
|  | +612 | +T | 147/6753 | 2.2% | Insertion | Frameshift |
|  | +636 | +T | 330/6753 | 4.9% | Insertion | Frameshift |
| TEVs-PEST | - | - | - | - | - | - |
| Intergenic | +1,564 | +A | 965/6766 | 14.3% | Insertion | - |
| P | +1,669 | +A | 187/6769 | 2.8% | Insertion | Frameshift |

**SIR-C-HighTEVp-P6 bc5—bc8**

|  | Position | Variant | N (q>20) | Freq % | Mutation | Effect on CDS |
|---|---|---|---|---|---|---|
| Upstream N | –49 | +A | 578/6166 | 9.4% | Insertion | - |
|  | –21 | -N | 205/6166 | 3.3% | Deletion | - |
|  | –20 | +G | 298/6166 | 4.8% | Insertion | - |

*Table 2 continued on next page*

*Table 2 continued*

**SIR-C-HighTEVp-P6 bc5—bc8**

|  | Position | Variant | N (q>20) | Freq % | Mutation | Effect on CDS |
|---|---|---|---|---|---|---|
|  | −19 | A>G | 3305/5625 | 58.8% | Substitution | - |
| N gene | +108 | +A | 179/6166 | 2.9% | Insertion | Frameshift |
|  | +166 | +T | 171/6165 | 2.8% | Insertion | Frameshift |
|  | +237 | +T | 514/6164 | 8.3% | Insertion | Frameshift |
|  | +466 | +A | 158/6166 | 2.6% | Insertion | Frameshift |
|  | +636 | +T | 318/6170 | 5.2% | Insertion | Frameshift |
| TEVs-PEST | +1,357 | G>T | 436/5995 | 7.3% | Substitution | Missense G453X |
| Intergenic | +1,564 | +A | 1019/6184 | 16.5% | Insertion | - |
| P | +1,669 | +A | 165/6185 | 2.7% | Insertion | Frameshift |

**SIR-D-HighTEVp-P6 bc5—bc9**

|  | Position | Variant | N (q>20) | Freq % | Mutation | Effect on CDS |
|---|---|---|---|---|---|---|
| Upstream N | −49 | +A | 562/6355 | 8.8% | Insertion | - |
|  | −21 | -N | 228/6356 | 3.6% | Deletion | - |
|  | −20 | +G | 314/6356 | 4.9% | Insertion | - |
|  | −19 | A>G | 2816/5789 | 48.6% | Substitution | - |
|  | -9 | A>T | 139/6104 | 2.3% | Substitution | - |
|  | -6 | C>T | 176/6275 | 2.8% | Substitution | - |
|  | -5 | C>A | 121/5995 | 2.0% | Substitution | - |
| N gene | +108 | +A | 175/6357 | 2.8% | Insertion | Frameshift |
|  | +237 | +T | 474/6358 | 7.5% | Insertion | Frameshift |
|  | +245 | +G | 131/6358 | 2.1% | Insertion | Frameshift |
|  | +466 | +A | 167/6359 | 2.6% | Insertion | Frameshift |
|  | +636 | +T | 316/6360 | 5.0% | Insertion | Frameshift |
| TEVs-PEST | - | - | - | - | - | - |
| Intergenic | +1,564 | +A | 947/6365 | 14.9% | Insertion | - |
| P | +1,669 | +A | 139/6365 | 2.2% | Insertion | Frameshift |

**SIR-A-LowTEVp-P6 bc4—bc5**

|  | Position | Variant | N (q>20) | Freq % | Mutation | Effect on CDS |
|---|---|---|---|---|---|---|
| Upstream N | −49 | +A | 588/6703 | 8.8% | Insertion | - |
|  | −19 | A>G | 369/6525 | 5.7% | Substitution | - |
| N gene | +108 | +A | 259/6704 | 3.9% | Insertion | Frameshift |
|  | +166 | +T | 173/6704 | 2.6% | Insertion | Frameshift |
|  | +237 | +T | 584/6703 | 8.7% | Insertion | Frameshift |
|  | +246 | +G | 145/6703 | 2.2% | Insertion | Frameshift |
|  | +466 | +A | 196/6704 | 2.9% | Insertion | Frameshift |
|  | +636 | +T | 366/6705 | 5.5% | Insertion | Frameshift |
| TEVs-PEST | +1,357 | G>T | 681/6468 | 10.5% | Substitution | Missense G453X |
| Intergenic | +1,564 | +A | 1035/6711 | 15.4% | Insertion | - |
| P | +1,669 | +A | 161/6711 | 2.4% | Insertion | Frameshift |

*Table 2 continued on next page*

*Table 2 continued*

**SIR-B-LowTEVp-P6 bc4—bc6**

|  | Position | Variant | N (q>20) | Freq % | Mutation | Effect on CDS |
|---|---|---|---|---|---|---|
| Upstream N | –49 | +A | 550/6112 | 9.0% | Insertion | - |
|  | –19 | A>G | 317/5985 | 5.3% | Substitution | - |
| N gene | +108 | +A | 186/6117 | 3.0% | Insertion | Frameshift |
|  | +166 | +T | 131/6117 | 2.1% | Insertion | Frameshift |
|  | +237 | +T | 486/6116 | 7.9% | Insertion | Frameshift |
|  | +466 | +A | 148/6118 | 2.4% | Insertion | Frameshift |
|  | +612 | +T | 125/6120 | 2.0% | Insertion | Frameshift |
|  | +636 | +T | 303/6119 | 5.0% | Insertion | Frameshift |
| TEVs-PEST | +1,357 | G>T | 360/5983 | 6.0% | Substitution | Missense G453X |
| Intergenic | +1,564 | +A | 946/6133 | 15.4% | Insertion | - |
| P | +1,669 | +A | 138/6133 | 2.3% | Insertion | Frameshift |

**SIR-C-LowTEVp-P6 bc4—bc7**

|  | Position | Variant | N (q>20) | Freq % | Mutation | Effect on CDS |
|---|---|---|---|---|---|---|
| Upstream N | –49 | +A | 494/5209 | 9.5% | Insertion | - |
|  | –20 | +G | 123/5209 | 2.4% | Insertion | - |
|  | –19 | A>G | 2864/4984 | 5.7% | Substitution | - |
| N gene | +108 | +A | 167/5210 | 3.2% | Insertion | Frameshift |
|  | +166 | +T | 136/5210 | 2.6% | Insertion | Frameshift |
|  | +237 | +T | 400/5210 | 7.7% | Insertion | Frameshift |
|  | +245 | +G | 123/5210 | 2.4% | Insertion | Frameshift |
|  | +466 | +A | 146/5213 | 2.8% | Insertion | Frameshift |
|  | +636 | +T | 261/5214 | 5.0% | Insertion | Frameshift |
| TEVs-PEST | +1,357 | G>T | 546/5066 | 10.8% | Substitution | Missense G453X |
| Intergenic | +1,564 | +A | 816/5212 | 15.7% | Insertion | - |
| P | +1,669 | +A | 120/5212 | 2.3% | Insertion | Frameshift |

**SIR-D-LowTEVp-P6 bc4—bc7**

|  | Position | Variant | N (q>20) | Freq % | Mutation | Effect on CDS |
|---|---|---|---|---|---|---|
| Upstream N | –49 | +A | 492/5279 | 9.3% | Insertion | - |
|  | –21 | -N | 114/5279 | 2.2% | Deletion | - |
|  | –20 | +G | 119/5279 | 2.3% | Insertion | - |
|  | –19 | A>G | 1553/5049 | 30.8% | Substitution | - |
|  | -9 | A>T | 104/5189 | 2.0% | Substitution | - |
| N gene | +108 | +A | 163/5279 | 3.1% | Insertion | Frameshift |
|  | +166 | +T | 129/5279 | 2.4% | Insertion | Frameshift |
|  | +237 | +T | 434/5279 | 8.2% | Insertion | Frameshift |
|  | +245 | +G | 106/5279 | 2.0% | Insertion | Frameshift |
|  | +466 | +A | 148/5281 | 2.8% | Insertion | Frameshift |
|  | +612 | +T | 120/5281 | 2.3% | Insertion | Frameshift |
|  | +636 | +T | 279/5281 | 5.3% | Insertion | Frameshift |
| TEVs-PEST | +1,357 | - | - | - | - | - |

*Table 2 continued*

**SIR-D-LowTEVp-P6 bc4—bc7**

| | Position | Variant | N (q>20) | Freq % | Mutation | Effect on CDS |
|---|---|---|---|---|---|---|
| Intergenic | +1,564 | +A | 831/5281 | 15.7% | Insertion | - |
| P | +1,669 | +A | 123/5281 | 2.3% | Insertion | Frameshift |

**SIR-A-HighTEVp-P8 bc6—bc7**

| | Position | Variant | N (q>20) | Freq % | Mutation | Effect on CDS |
|---|---|---|---|---|---|---|
| Upstream N | –49 | +A | 541/6868 | 7.9% | Insertion | - |
| | –21 | -N | 299/6868 | 4.4% | Deletion | - |
| | –20 | +G | 431/6868 | 6.3% | Insertion | - |
| | –19 | A>G | 3684/6150 | 60.0% | Substitution | - |
| N gene | +108 | +A | 198/6867 | 2.9% | Insertion | Frameshift |
| | +166 | +T | 157/6867 | 2.3% | Insertion | Frameshift |
| | +237 | +T | 583/6867 | 8.5% | Insertion | Frameshift |
| | +245 | +G | 138/6867 | 2.0% | Insertion | Frameshift |
| | +466 | +A | 181/6868 | 2.6% | Insertion | Frameshift |
| | +636 | +T | 342/6870 | 5.0% | Insertion | Frameshift |
| TEVs-PEST | +1,357 | G>T | 651/6620 | 9.8% | Substitution | Missense G453X |
| Intergenic | +1,564 | +A | 952/6896 | 13.8% | Insertion | - |
| P | +1,669 | +A | 144/6898 | 2.1% | Insertion | Frameshift |

**SIR-B-HighTEVp-P8 bc6—bc8**

| | Position | Variant | N (q>20) | Freq % | Mutation | Effect on CDS |
|---|---|---|---|---|---|---|
| Upstream N | –49 | +A | 571/6246 | 9.1% | Insertion | - |
| | –21 | -N | 182/6246 | 2.9% | Deletion | - |
| | –20 | +G | 319/6246 | 5.1% | Insertion | - |
| | –19 | A>G | 3836/5763 | 66.6% | Substitution | - |
| | –18 | A>C | 171/5940 | 2.9% | Substitution | - |
| N gene | +108 | +A | 197/6247 | 3.2% | Insertion | Frameshift |
| | +166 | +T | 167/6247 | 2.7% | Insertion | Frameshift |
| | +237 | +T | 486/6247 | 7.8% | Insertion | Frameshift |
| | +245 | +G | 145/6248 | 2.3% | Insertion | Frameshift |
| | +466 | +A | 149/6249 | 2.4% | Insertion | Frameshift |
| | +636 | +T | 323/6251 | 5.2% | Insertion | Frameshift |
| TEVs-PEST | +1,357 | G>T | 365/6068 | 6.0% | Substitution | Missense G453X |
| Intergenic | +1,564 | +A | 927/6259 | 14.8% | Insertion | - |
| P | +1,669 | +A | 152/6259 | 2.4% | Insertion | Frameshift |

**SIR-C-HighTEVp-P8 bc6—bc9**

| | Position | Variant | N (q>20) | Freq % | Mutation | Effect on CDS |
|---|---|---|---|---|---|---|
| Upstream N | –49 | +A | 598/6403 | 9.3% | Insertion | - |
| | –19 | A>G | 6024/6304 | 95.6% | Substitution | - |
| N gene | +108 | +A | 200/6404 | 3.1% | Insertion | Frameshift |
| | +166 | +T | 146/6404 | 2.3% | Insertion | Frameshift |
| | +237 | +T | 518/6405 | 8.1% | Insertion | Frameshift |

*Table 2 continued on next page*

*Table 2 continued*

**SIR-C-HighTEVp-P8 bc6—bc9**

|  | Position | Variant | N (q>20) | Freq % | Mutation | Effect on CDS |
|---|---|---|---|---|---|---|
|  | +245 | +G | 158/6405 | 2.5% | Insertion | Frameshift |
|  | +466 | +A | 172/6406 | 2.7% | Insertion | Frameshift |
|  | +636 | +T | 311/6407 | 4.9% | Insertion | Frameshift |
| TEVs-PEST | - | - | - | - | - | - |
| Intergenic | +1,564 | +A | 986/6410 | 15.4% | Insertion | - |
| P | +1,669 | +A | 139/6408 | 2.2% | Insertion | Frameshift |

**SIR-D-HighTEVp-P8 bc6—bc10**

|  | Position | Variant | N (q>20) | Freq % | Mutation | Effect on CDS |
|---|---|---|---|---|---|---|
| Upstream N | –49 | +A | 482/5760 | 8.4% | Insertion | - |
|  | –19 | A>G | 5092/5625 | 9.1% | Substitution | - |
|  | –18 | A>G | 155/5609 | 2.8% | Substitution | - |
|  | -9 | A>T | 247/5402 | 4.6% | Substitution | - |
|  | -9 | A>G | 449/5402 | 8.3% | Substitution | - |
|  | -9 | +G | 120/5761 | 2.1% | Insertion | - |
|  | -6 | C>T | 680/5586 | 12.2% | Substitution | - |
|  | -6 | +T | 167/5761 | 2.9% | Insertion | - |
|  | -5 | C>A | 153/5412 | 2.8% | Substitution | - |
| N gene | +108 | +A | 163/5763 | 2.8% | Insertion | Frameshift |
|  | +166 | +T | 119/5763 | 2.1% | Insertion | Frameshift |
|  | +237 | +T | 414/5763 | 7.2% | Insertion | Frameshift |
|  | +466 | +A | 119/5764 | 2.1% | Insertion | Frameshift |
|  | +612 | +T | 127/5764 | 2.2% | Insertion | Frameshift |
|  | +636 | +T | 291/5764 | 5.0% | Insertion | Frameshift |
| TEVs-PEST | - | - | - | - | - | - |
| Intergenic | +1,564 | +A | 861/5766 | 14.9% | Insertion | - |
| P | +1,669 | +A | 137/5766 | 2.4% | Insertion | Frameshift |

**SIR-A-LowTEVp-P8 bc4—bc9**

|  | Position | Variant | N (q>20) | Freq % | Mutation | Effect on CDS |
|---|---|---|---|---|---|---|
| Upstream N | –49 | +A | 646/7058 | 9.2% | Insertion | - |
|  | –21 | -N | 252/7059 | 3.6% | Deletion | - |
|  | –20 | +G | 417/7059 | 5.9% | Insertion | - |
|  | –19 | A>G | 2752/6358 | 43.3% | Substitution | - |
|  | -6 | C>T | 171/6942 | 2.5% | Substitution | - |
|  | -5 | C>A | 542/6530 | 8.3% | Substitution | - |
| N gene | +108 | +A | 346/7058 | 4.9% | Insertion | Frameshift |
|  | +166 | +T | 178/7058 | 2.5% | Insertion | Frameshift |
|  | +237 | +T | 622/7058 | 8.8% | Insertion | Frameshift |
|  | +245 | +G | 161/7058 | 2.3% | Insertion | Frameshift |
|  | +466 | +A | 194/7059 | 2.7% | Insertion | Frameshift |

*Table 2 continued on next page*

*Table 2 continued*

**SIR-A-LowTEVp-P8 bc4—bc9**

|  | Position | Variant | N (q>20) | Freq % | Mutation | Effect on CDS |
|---|---|---|---|---|---|---|
|  | +612 | +T | 150/7060 | 2.1% | Insertion | Frameshift |
|  | +636 | +T | 345/7060 | 4.9% | Insertion | Frameshift |
|  | +795 | T>C | 1604/6265 | 25.6% | Substitution | Silent F265 |
|  | +795 | +C | 318/7061 | 4.5% | Insertion | Frameshift |
| TEVs-PEST | +1,357 | G>T | 1122/6684 | 16.8% | Substitution | Missense G453X |
| Intergenic | +1,564 | +A | 1079/7085 | 15.2% | Insertion | - |
| P | +1,669 | +A | 161/7090 | 2.3% | Insertion | Frameshift |

**SIR-B-LowTEVp-P8 bc4—bc10**

|  | Position | Variant | N (q>20) | Freq % | Mutation | Effect on CDS |
|---|---|---|---|---|---|---|
| Upstream N | –49 | +A | 647/6759 | 9.6% | Insertion | - |
|  | –21 | -N | 242/6761 | 3.6% | Deletion | - |
|  | –20 | +G | 371/6761 | 5.5% | Insertion | - |
|  | –19 | A>G | 2200/6168 | 35.7% | Substitution | - |
|  | –18 | A>C | 400/6309 | 6.3% | Substitution | - |
| N gene | +108 | +A | 224/6761 | 3.3% | Insertion | Frameshift |
|  | +166 | +T | 157/6761 | 2.3% | Insertion | Frameshift |
|  | +237 | +T | 575/6760 | 8.5% | Insertion | Frameshift |
|  | +466 | +A | 189/6764 | 2.8% | Insertion | Frameshift |
|  | +636 | +T | 353/6763 | 5.2% | Insertion | Frameshift |
|  | +1,349 | C>A | 144/6671 | 2.2% | Substitution | Missense S450X |
| TEVs-PEST | +1,357 | G>T | 1192/6372 | 18.7% | Substitution | Missense G453X |
| Intergenic | +1,564 | +A | 1026/6769 | 15.2% | Insertion | - |
| P | +1,669 | +A | 173/6772 | 2.6% | Insertion | Frameshift |

**SIR-C-LowTEVp-P8 bc4—bc11**

|  | Position | Variant | N (q>20) | Freq % | Mutation | Effect on CDS |
|---|---|---|---|---|---|---|
| Upstream N | –49 | +A | 614/6893 | 8.9% | Insertion | - |
|  | –20 | +G | 261/6893 | 3.8% | Insertion | - |
|  | –19 | A>G | 5317/6466 | 82.2% | Substitution | - |
| N gene | +108 | +A | 215/6894 | 3.1% | Insertion | Frameshift |
|  | +237 | +T | 564/6894 | 8.2% | Insertion | Frameshift |
|  | +466 | +A | 207/6895 | 3.0% | Insertion | Frameshift |
|  | +636 | +T | 364/6895 | 5.3% | Insertion | Frameshift |
| TEVs-PEST | +1,357 | G>T | 1013/6551 | 15.5% | Substitution | Missense G453X |
| Intergenic | +1,564 | +A | 1053/6920 | 15.2% | Insertion | - |
| P | - | - | - | - | - | - |

**SIR-D-LowTEVp-P8 bc4—bc12**

|  | Position | Variant | N (q>20) | Freq % | Mutation | Effect on CDS |
|---|---|---|---|---|---|---|
| Upstream N | –49 | +A | 541/5872 | 9.2% | Insertion | - |
|  | –20 | +G | 190/5872 | 3.2% | Insertion | - |
|  | –19 | A>G | 4259/5565 | 76.5% | Substitution | - |

*Table 2 continued on next page*

*Table 2 continued*

**SIR-D-LowTEVp-P8 bc4—bc12**

| | Position | Variant | N (q>20) | Freq % | Mutation | Effect on CDS |
|---|---|---|---|---|---|---|
| | -9 | A>T | 141/5738 | 2.5% | Substitution | - |
| N gene | +108 | +A | 168/5876 | 2.9% | Insertion | Frameshift |
| | +166 | +T | 154/5876 | 2.6% | Insertion | Frameshift |
| | +237 | +T | 491/5876 | 8.4% | Insertion | Frameshift |
| | +245 | +G | 133/5876 | 2.3% | Insertion | Frameshift |
| | +332 | +A | 123/5876 | 2.1% | Insertion | Frameshift |
| | +466 | +A | 152/5876 | 2.6% | Insertion | Frameshift |
| | +612 | +T | 134/5876 | 2.3% | Insertion | Frameshift |
| | +636 | +T | 324/5876 | 5.5% | Insertion | Frameshift |
| TEVs-PEST | +1,357 | G>T | 521/5707 | 9.1% | Substitution | Missense G453X |
| Intergenic | +1,564 | +A | 996/5881 | 17.0% | Insertion | - |
| P | +1,669 | +A | 150/5882 | 2.6% | Insertion | Frameshift |

we engineered each of the two nonsense mutations previously reported (*Matsuyama et al., 2019*) (d.C1349G and d.G1357T, leading to stop insertion at S450 and G453, respectively; *Figure 2F*) in the SiR cDNA, generating two viruses named SiR-S450X and SiR-G453X (*Figure 3A*, *Figure 3—figure supplement 1*). First, we confirmed the loss of functional TEVs in the PEST linker in the engineered-revertants by observing the TEVp-dependent virally driven GFP expression in vitro (*Figure 3—figure supplement 1*). Next, we assessed the in vivo cytotoxicity of SiR, SiR-G453X and ΔG-Rab expressing CRE by injecting them in the CA1 hippocampal region of CRE-dependent tdTomato reporter mice (*Rosa26^LSL-tdTomato*) and analysing the number of infected neurons at different time points post injection (p.i.) as in our previous study (*Ciabatti et al., 2017*; *Figure 3B*). We detected no decrease of tdTomato⁺ neurons in SiR-infected hippocampi (4109±266 tdTomato +neurons at 1 week p.i.; 4458±739 tdTomato +neurons at 2 months p.i.; one-way ANOVA, $F$=0.08, p=0.92, *Figure 3C–D*) while only 44% of tdTomato +neurons were detected in Rabies-targeted and 60% in SiR-G453X-targeted hippocampi at 2 months p.i. (1422±184 at 1 week versus 624±114 at 2 months p.i. for ΔG-Rab; one-way ANOVA, $F$=11.55, p=0.003; 3052+508 at 1 week versus 1829+198 at 2 months p.i. for SiR-G453X; one-way ANOVA, $F$=4.27, p=0.05; *Figure 3C–D*). Additionally, we confirmed inactivation of revertant-free SiR by analysing the decrease of Rabies transcripts in the infected hippocampi over times (*Figure 3—figure supplement 2*). These results support the lack of toxicity of SiR on the infected neurons, in line with our previous findings (*Ciabatti et al., 2017*). Moreover, these data confirm the requirement for an intact PEST sequence to sustain the self-inactivating behaviour of SiR and suggest that PEST-targeting mutations do not occur in vivo. Notably, a fraction of tdTomato +neurons survived in ΔG-Rab-CRE-injected brains, differing from what we observed when injecting ΔG-Rab-GFP, where no cells were detected at 3 weeks p.i. (*Figure 3C–D*; *Ciabatti et al., 2017*). To experimentally confirm that revertant particles indeed do not emerge in vivo during long-term SiR experiments, we prepared NGS libraries of SiR genomes extracted from hippocampi of injected animals before SiR switch off and sequenced them by SMRT sequencing (*Figure 3E* and *Figure 2—figure supplement 2*). In all three independent experiments, no revertant mutations had accumulated in vivo above threshold prior to the switching off of the virus (*Figure 3F*, *Table 3*).

To further confirm the lack of any toxic effect in SiR-targeted neurons we also performed longitudinal imaging of cortical neurons using 2-photon microscopy. These longitudinal experiments allowed us to follow the morphology and survival of the same identified SiRtargeted neurons over time in living mice, thereby giving more direct evidence of the potential cytotoxicity or lack thereof associated with SiR. We imaged SiR-CRE or ΔG-Rab-CRE labelled neurons in the cerebral cortex of *Rosa26^LSL-tdTomato* mice for up to 5 months p.i. (*Figure 4A–B*). The total number of detectable tdTomato⁺ neurons increased in SiR injected animals between 1 and 2 weeks and remained constant for the entire duration of the experiment (*Figure 4B*), while ΔG-Rab–injected cortices show a decrease of total number of

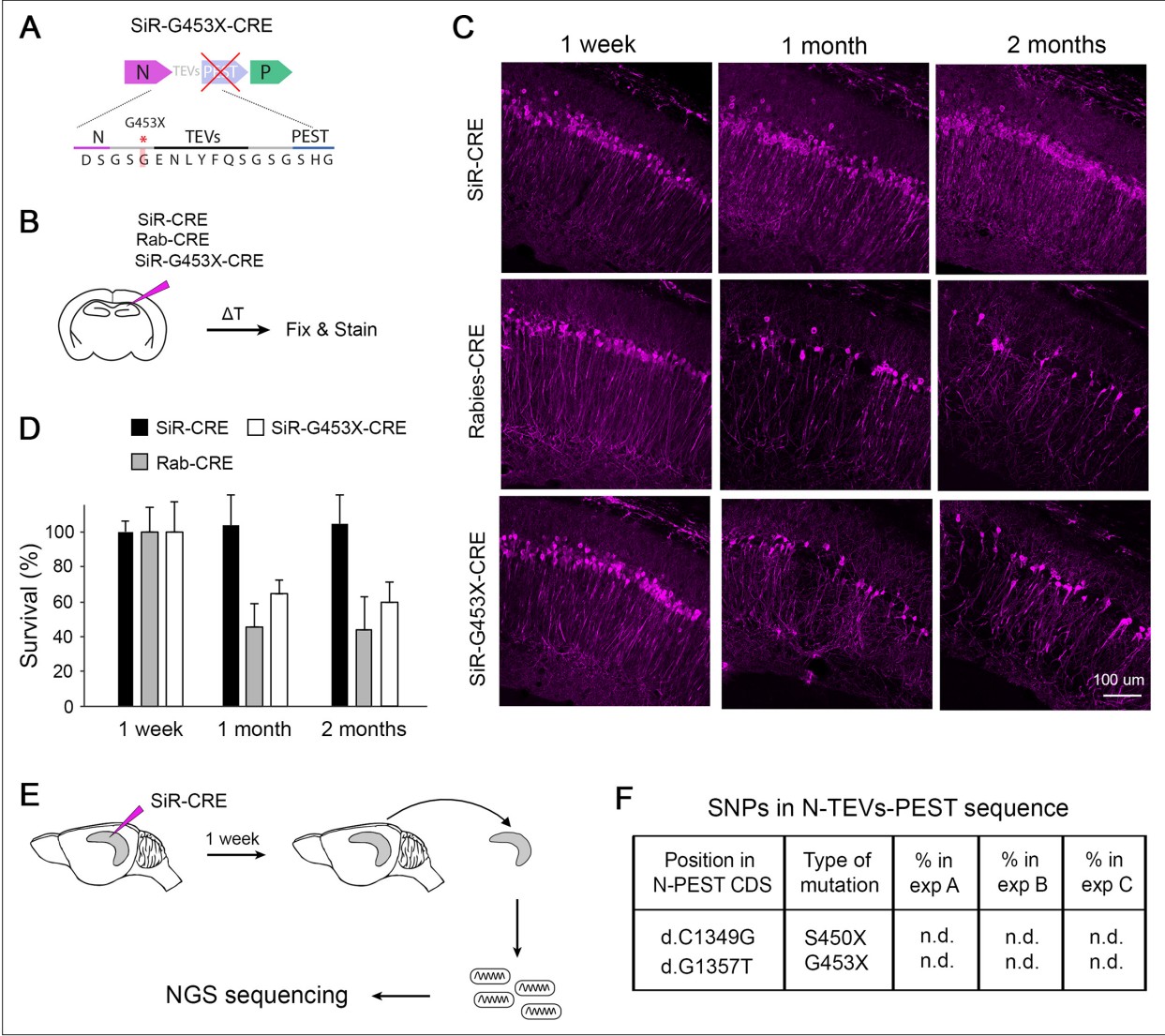

**Figure 3.** Revertant-free SiR, but not PEST-mutant, is non-toxic and does not accumulate PEST-targeting mutations in vivo. (**A**) Scheme of the engineered PEST-mutant SiR (SiR-G453X). (**B**) Experimental procedure. (**C**) Confocal images of hippocampal sections of *Rosa26^LSL-tdTomato* mice infected with SiR-CRE, Rab-CRE, SiR-G453X and imaged at 1 week, 1 month and 2 months p.i. Scale bar, 50 μm. (**D**) Number of tdTomato positive neurons at 1 week, 1 months, and 2 months p.i. normalized to 1 week time point (mean ± SEM, n=4 animals per virus per time point). (**E**) Experimental procedure for the sequencing of SiR particles from injected hippocampi at 1 week p.i. (**F**) List of PEST-inactivating mutations above 2% thresholds with relative frequency in each animal (n.d. indicates that the mutation was not detected above threshold; n=3 animals).

The online version of this article includes the following source data and figure supplement(s) for figure 3:

**Source data 1.** tdTomato⁺ positive neurons in injected Hippocampi with Rab, SiR or Pest-mutant SiR.

**Figure supplement 1.** SiR revertants lose functional TEVs and PEST domain.

**Figure supplement 2.** SiR RNA in injected hippocampi.

tdTomato⁺ neurons over time (***Figure 4B***). Importantly, nearly all the SiR-targeted neurons imaged at 1 week were detected in subsequent imaging sessions (97%±1 tdTomato⁺ at 21 weeks p.i.; ***Figure 4C***) in contrast to ΔG-Rab-infected neurons, where ~70% of the neurons detected at 1 week had died by 9 weeks p.i. (29%±2 tdTomato⁺ at 21 weeks; ***Figure 4C***). These results show virtually no loss of SiR-labelled neurons during the entire imaging period (5 months) and confirm the lack of any observable cytotoxic effect of SiR on the recipient neurons (***Figure 4B–D*** and ***Figure 4—figure supplement 1***).

**Table 3.** List of detected mutations above 2% threshold in purified SiR viruses recovered from injected hippocampi sequenced by SMRT NGS sequencing.

The position of the mutations is defined considering +1 the first base of the nucleoprotein N coding sequence.

**NGS sequencing results of purified viruses used in vivo**

**SIR-CRE purified bc3—bc5**

|  | Position | Variant | N (q>20) | Freq % | Mutation | Effect on CDS |
|---|---|---|---|---|---|---|
| Upstream N | –49 | +A | 238/5196 | 4.6% | Insertion | - |
| N gene | +237 | +T | 199/5196 | 3.8% | Insertion | Frameshift |
|  | +636 | +T | 150/5200 | 2.9% | Insertion | Frameshift |
| TEVs-PEST | - | - | - | - | - | - |
| Intergenic | +1,564 | +A | 544/5205 | 10.5% | Insertion | - |
| P | - | - | - | - | - | - |

**SIR-CRE purified, 1 week p.i. in vivo (A) bc5—bc10**

|  | Position | Variant | N (q>20) | Freq % | Mutation | Effect on CDS |
|---|---|---|---|---|---|---|
| Upstream N | –49 | +A | 474/5211 | 9.1% | Insertion | - |
|  | –21 | +A | 110/5211 | 2.1% | Insertion | - |
| N gene | +108 | +A | 176/5211 | 3.4% | Insertion | Frameshift |
|  | +166 | +T | 132/5211 | 2.5% | Insertion | Frameshift |
|  | +237 | +T | 389/5211 | 7.5% | Insertion | Frameshift |
|  | +245 | +G | 108/5211 | 2.1% | Insertion | Frameshift |
|  | +466 | +A | 135/5211 | 2.6% | Insertion | Frameshift |
|  | +612 | +T | 108/5210 | 2.1% | Insertion | Frameshift |
|  | +636 | +T | 288/5210 | 5.5% | Insertion | Frameshift |
| TEVs-PEST | - | - | - | - | - | - |
| Intergenic | +1,564 | +A | 773/5213 | 14.8% | Insertion | - |
| P | +1,669 | +A | 128/5213 | 2.5% | Insertion | Frameshift |

**SIR-CRE purified, 1 week p.i. in vivo (B) bc5—bc11**

|  | Position | Variant | N (q>20) | Freq % | Mutation | Effect on CDS |
|---|---|---|---|---|---|---|
| Upstream N | –49 | +A | 482/5542 | 8.7% | Insertion | - |
| N gene | +108 | +A | 157/5543 | 2.8% | Insertion | Frameshift |
|  | +166 | +T | 125/5543 | 2.3% | Insertion | Frameshift |
|  | +237 | +T | 402/5543 | 7.3% | Insertion | Frameshift |
|  | +245 | +G | 123/5543 | 2.2% | Insertion | Frameshift |
|  | +466 | +A | 157/5543 | 2.8% | Insertion | Frameshift |
|  | +612 | +T | 112/5543 | 2.0% | Insertion | Frameshift |
|  | +636 | +T | 276/5543 | 5.0% | Insertion | Frameshift |
| TEVs-PEST | - | - | - | - | - | - |
| Intergenic | +1,564 | +A | 744/5542 | 13.4% | Insertion | - |
| P | +1,669 | +A | 144/5542 | 2.6% | Insertion | Frameshift |

**SIR-CRE purified, 1 week p.i. in vivo (C) bc5—bc12**

|  | Position | Variant | N (q>20) | Freq % | Mutation | Effect on CDS |
|---|---|---|---|---|---|---|
| Upstream N | –49 | +A | 481/5150 | 9.3% | Insertion | - |

*Table 3 continued on next page*

*Table 3 continued*

**SIR-CRE purified, 1 week p.i. in vivo (C) bc5—bc12**

| | Position | Variant | N (q>20) | Freq % | Mutation | Effect on CDS |
|---|---|---|---|---|---|---|
| N gene | +108 | +A | 137/5150 | 2.7% | Insertion | Frameshift |
| | +166 | +T | 118/5150 | 2.3% | Insertion | Frameshift |
| | +237 | +T | 390/5150 | 7.6% | Insertion | Frameshift |
| | +245 | +G | 104/5150 | 2.0% | Insertion | Frameshift |
| | +466 | +A | 140/5150 | 2.7% | Insertion | Frameshift |
| | +612 | +T | 116/5150 | 2.3% | Insertion | Frameshift |
| | +636 | +T | 255/5150 | 5.0% | Insertion | Frameshift |
| TEVs-PEST | - | - | - | - | - | - |
| Intergenic | +1,564 | +A | 739/5148 | 14.4% | Insertion | - |
| P | +1,669 | +A | 130/5148 | 2.5% | Insertion | Frameshift |

**SIR-G453X-CRE purified bc3—bc11**

| | Position | Variant | N (q>20) | Freq % | Mutation | Effect on CDS |
|---|---|---|---|---|---|---|
| Upstream N | –49 | +A | 211/4886 | 4.3% | Insertion | - |
| N gene | +237 | +T | 244/4890 | 5.0% | Insertion | Frameshift |
| | +636 | +T | 138/4911 | 2.8% | Insertion | Frameshift |
| TEVs-PEST | +1,357 | G>T | 4780/4912 | 97.3% | Substitution | Missense G453X |
| Intergenic | +1,564 | +A | 502/4924 | 10.2% | Insertion | - |
| P | - | - | - | - | - | - |

## SiR transsynaptic spreading

We then tested the ability of revertant-free SiR to trace neural circuits transsynaptically in the mouse brain. ΔG-Rabies vectors can be pseudotyped with the chimeric EnvA glycoprotein to selectively infect neurons expressing the TVA receptor, which is not endogenously expressed by mammalian cells (*Wickersham et al., 2007b*). We injected the nucleus accumbens (NAc) of CRE-dependent tdTomato reporter mice with an AAV expressing either TVA and the rabies G or TVA only. After 3 weeks, we re-injected the NAc with EnvA-pseudotyped revertant-free SiR-CRE or EnvA-pseudotyped SiR-G453X-CRE and assessed the CRE-dependent tdTomato expression presynaptically, in the basolateral amygdala (BLA). At 1 month post SiR injection, we detected no tdTomato[+] cells in the BLA in TVA-only-injected animals, confirming the G-dependency for SiR transsynaptic spreading (*Figure 5B–C*). In contrast, as expected, transsynaptic spreading was apparent in the TVA +G condition. We observed similar numbers of presynaptically traced neurons in both SiR-CRE and SiR-G453X-CRE injected brains (169±24 and 190±36 tdTomato[+] neurons, respectively; two-tailed t-test, p=0.64; *Figure 5B–C*). However, tdTomato[+] microglial cells were only detected in the SiR-G453X-CRE condition indicating the re-emergence of toxicity of the revertant mutants (*Figure 5B*). We also tested the effect of supplying TEV protease to the starting cells, as this has been suggested to be a necessary step to ensure trans-synapitc spreading. While the previous experiments unambiguously show that TEVp is not necessary for the transsynaptic spreading of SiR, the injection of an AAV expressing TEVp in the NAc did lead to an increase in the number of transsynaptically labelled BLA neurons (366±69 tdTomato[+] neurons; two-tailed t-test, *P*=0.04; *Figure 5C*), indicating that TEVp-dependent SiR reactivation in starter cells can improve its spreading (*Jin et al., 2023*).

We recently showed that a novel SiR-N2c vector, derived from the neurotropic CVS-N2c Rabies strain, displays enhanced transsynaptic spreading and improved peripheral neurotropism over the original SAD B19-derived SiR (*Lee et al., 2023*). Hence, for completeness, we compared the transsynaptic spreading efficacty of EnvA-pseudotyped revertant-free SiR-N2c and the original SiR. SiR-N2c labelled a greater number of BLA neurons at 1 month p.i. than what was detected with SiR (1691 ±

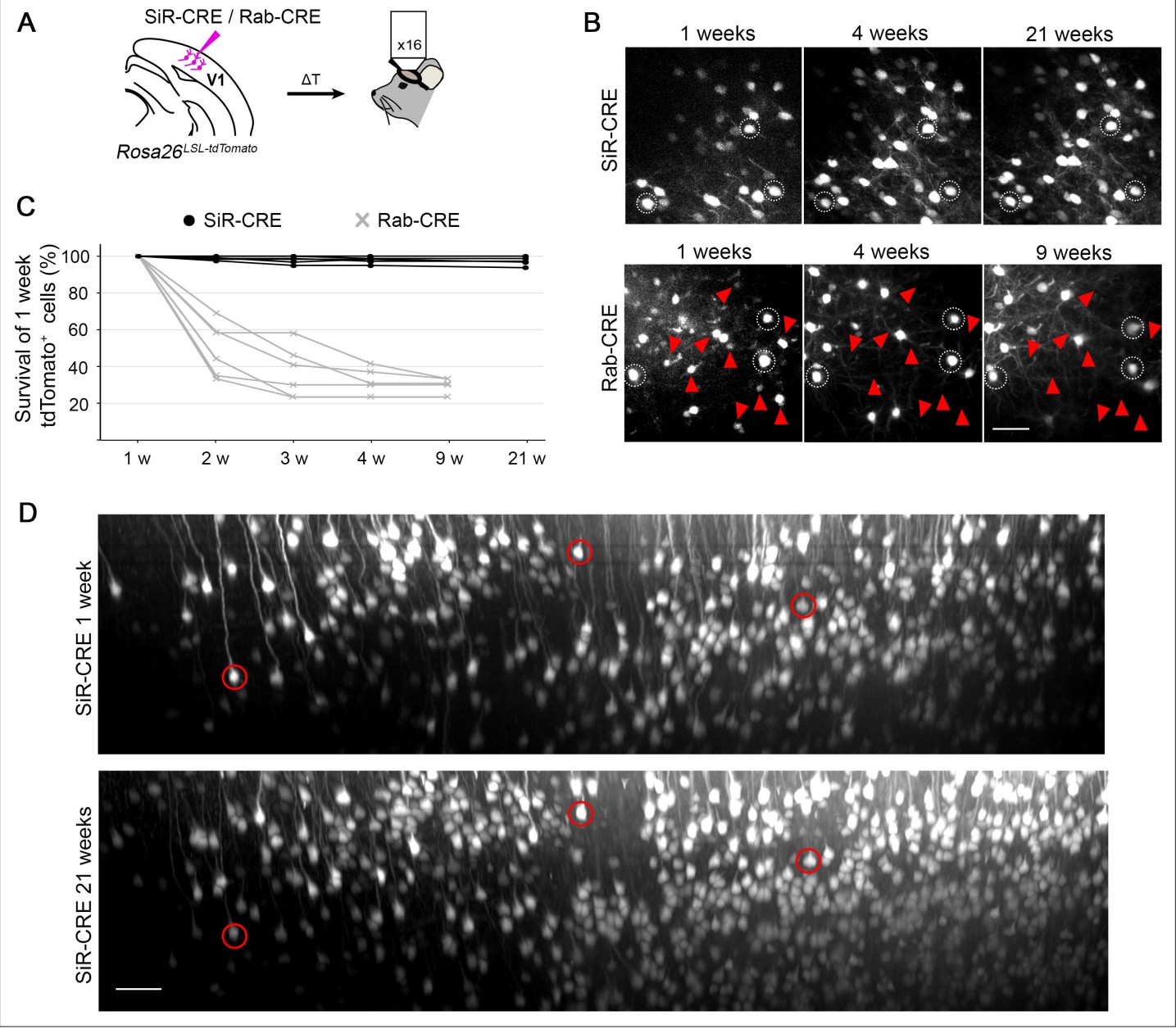

**Figure 4.** 2-photon in vivo longitudinal imaging of revertant-free SiR-infected cortical neurons reveals no toxicity and unaltered neuronal morphology after 5 months. (**A**) Schematic of SiR-CRE or Rab-CRE injection in *Rosa26^LSL-tdTomato* mice in V1 followed by in vivo imaging. (**B**) Two-photon maximal projection of the same field in SiR-CRE and RabCRE injected cortices at 1, 4, and 21 weeks p.i. or 1, 4, and 9 weeks, respectively. Red arrowheads mark tdTomato positive neurons detected at 1 week that disappear in later recordings. Scale bar 50 μm. (**C**) Survival of the tdTomato-positive cells recorded at 1 week over time. (ROIs = 6 per virus. n=2 animals per virus). (**D**) Two-photon maximal projection of the same large field in SiR-CRE injected cortices at 1 week and 21 weeks p.i. Scale bar 50 μm.

The online version of this article includes the following source data and figure supplement(s) for figure 4:

**Source data 1.** tdTomato⁺ positive neurons in injected cortices with Rab or SiR.

**Figure supplement 1.** Two-photon in vivo longitudinal imaging of revertant-free SiR-infected cortical neurons.

112 tdTomato⁺ neurons traced by SiR-N2c; two-tailed t-test, p=2 × 10⁵; *Figure 5D–E*). Additionally, TEVp expression in the starter cells in SiR-N2c tracing experiments had a negligible effect on the overall transsynaptic spreading (1934±135 tdTomato⁺ neurons traced by SiR-N2c in presence of TEVp; two-tailed t-test, p=0.24; *Figure 5D–E*). Since the use of G from the CVS-N2c Rabies strain (G_N2c) has been shown to improve ΔG-Rabies (SAD-B19) retrograde tracing (*Zhu et al., 2020*), we tested if

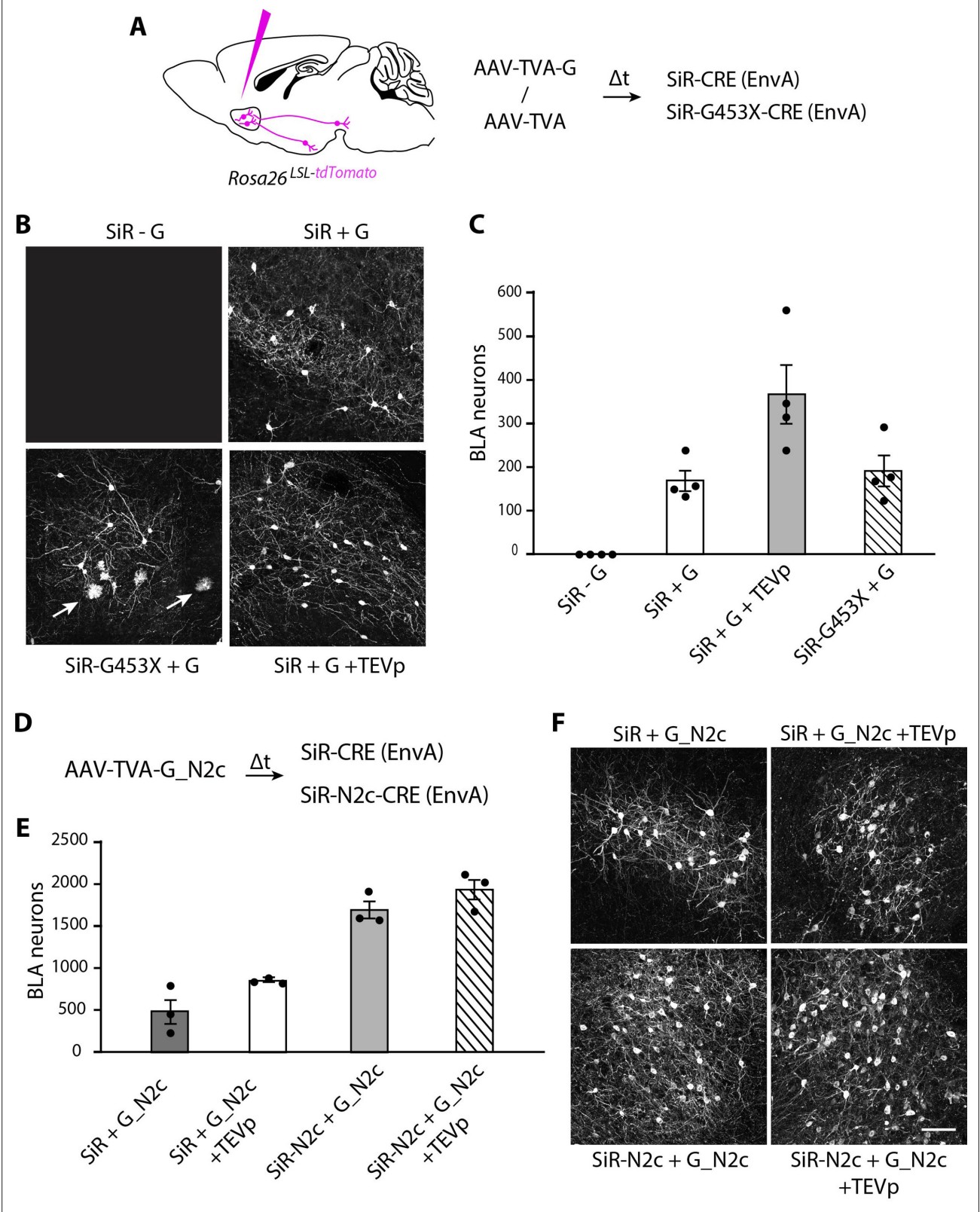

**Figure 5.** SiR vectors transsynaptic tracing of neural circuits in the central nervous system. (**A**) Experimental design for the transsynaptic tracing of NAc inputs using EnvA-pseudotyped SiR-CRE or SiR-G453X-CRE in *Rosa26$^{LSL-tdTomato}$* mice. (**B**) Confocal images of BLA area of *Rosa26$^{LSL-tdTomato}$* mice infected with SiR-CRE or SiR-G453X-CRE. Arrows point to tdTomato$^+$ microglia. (**C**) Number of tdTomato-positive neurons in the BLA at 1 month post

*Figure 5 continued on next page*

*Figure 5 continued*

SiR injection (mean ± SEM, n=4 animals per condition). (**E**) Number of tdTomato⁺ neurons in the BLA at 1 month post SiR injection (mean ± SEM, n=3 animals per condition). (**F**) Confocal images of BLA area of *Rosa26^LSL-tdTomato* mice infected with SiR-CRE or SiR-N2c-CRE. Scale bar, 100 µm.

The online version of this article includes the following source data for figure 5:

**Source data 1.** tdTomato⁺ positive BLA neurons upon transsynaptically tracing with SiR, Pest-mutant SiR or SiR-N2c.

complementing EnvA-pseudotyped SiR with G_N2c in the NAc could increase its spreading. While we detected more BLA tdTomato⁺ neurons than in our previous experiments, complementing SiR with G_N2c still labelled less neurons than SiR-N2c, even when TEVp was provided to the starter cells (487±164 and 844±14 tdTomato⁺ neurons traced by SiR in absence or presence of TEVp, respectively; *Figure 5D–E*).

## Discussion

The development of technologies to record and perturb the activity of neurons within neural circuits has been instrumental for the recent progress in systems neuroscience. ΔG-Rabies viruses have been transformative in the study of neural circuit organization in animal models, especially mammals. The recent generation of a non-toxic SiR vector has opened the door to the long-term functional dissection of neural networks. One concern regarding its widespread use has been the risk that mutations could emerge and compromise SiR preparations by reverting the SiR vector to canonical and cytotoxic ΔG-Rabies.

Here we have investigated the genomic stability of SiR and showed that PEST-targeting mutations are rare and do not accumulate when SiR is produced directly from cDNA as previously described. However, we show that revertant mutants can emerge if SiR is extensively amplified in vitro, particularly in cells expressing suboptimal levels of TEVp, where revertant mutants have a specific replication advantage. Nonetheless, we also show that when production utilises HEK-TGG packaging cells expressing high levels of TEVp, even 8 rounds of amplification in vitro do not lead to the accumulation of PEST-targeting mutations above 5%. Notably, we found that TEVp activity inevitably decreases after several passages of amplification of HEK-TTG. thus fresh low passage packaging cells should always be used to produce SiR preparations. Our results suggest that stock for packaging cells should be made within a couple of passage after selection is established, and then used freshly defrosted to produce SiR viruses (equivalent to P0 cells in *Figure 2B–C*). Similarly, SiR supernatant stocks should be made directly from cDNA transfection and amplified for a maximum of 2 passages (equivalent to SiR P0 in *Figure 2E*) before being used for large scale SiR productions.

Another important question is, when revertant-free SiR is produced and used for tracing experiments, can PEST-targeting mutations emerge in vivo? Here we show that revertant-free SiR-CRE efficiently infect neurons in vivo without toxicity in cortical and subcortical regions for several months p.i. Importantly, PEST-mutant SiR is as toxic as canonical ΔG-Rabies, indicating that an intact PEST sequence is essential for SiR non-toxic behaviour and suggesting that revertant mutants do not emerge during in vivo experiments. We confirmed this by sequencing the SiR viral particles isolated from in vivo experiments and found no PEST-targeting mutations. Thus, the short lifetime of the SiR in the infected neurons does not permit PEST mutations to emerge and accumulate in vivo before viral disappearance when revertant-free SiR preparations are used.

ΔG-Rabies vectors are powerful tools for the dissection of neural circuit organization thanks to their ability to spread retrogradely to synpatically-connected neurons. Here, we show that EnvA-pseudotyped revertant-free SiR vectors effectively spread transsynpatically in the mouse brain. Importantly, the co-delivery of an AAV expressing TEVp in addition to G increase the number of traced neurons in presynaptic areas, likely due to the TEVp-dependent reactivation of SiR in vivo (*Ciabatti et al., 2017*), in line with recent results (*Jin et al., 2023*). This should be considered when planning transsynaptic tracing experiments using SiR. To improve SiR spreading efficiency, further studies should investigate the use of inducible TEVp, as we previously showed (*Ciabatti et al., 2017*), that could maximise spreading efficiency while minimising possible side effects of prolonged protease expression.

Interestingly, we found that the recently developed SiR-N2c vector, generated by applying the same proteasome-targeting modification to the genome of the CVS-N2c ΔG-Rabies strain (*Lee et al.,*

*2023*), show a higher number of retrogradely labelled neurons compared to the original SiR (SAD-B19; *Figure 5*). Additionally, the co-delivery of TEVp had a smaller effect on the number of neurons transsynaptically traced by SiR-N2c. Interestingly, the gap in trassynaptic spreading efficacy between SiR (SAD-B19) and SiR-N2c could not be filled by complementing the SiR with the neurotropic G_N2c. This could be linked to a more efficient packaging of SiR-N2c by G_N2c (*Reardon et al., 2016*; *Sumser et al., 2022*) or by the particularly high speed of CVS-N2c strain propagation (~12 hr; *Callaway, 2008*; *Hoshi et al., 2005*). These results point to SiR-N2c as the vector of choice for transsynaptic experiments.

Although PEST-inactivating mutations can be prevented during production and do not accumulate in vivo, strategies to further reduce or entirely eliminate the risk of their appearance could simplify viral production in other laboratories and allow the use of SiR in sensitive applications, *e.g.* re-targeting the same starter cells multiple times. In our experiments only two specific revertant mutations were identified, single base substitutions that introduce a stop signal either at the last amino acid of N or in the linker prior to TEVs and PEST (d.C1349G and d.G1357T) which accounted for the large majority of revertant mutations found in *Matsuyama et al., 2019*. Future studies should focus on investigating if this and other potential hotspots in the SiR genome can be optimised to simplify the production of SiR.

## Methods
### Contact for Reagents and Resource Sharing
Further information and requests for resources and reagents should be directed to the corresponding author: Ernesto Ciabatti (ciabatti@mrc-lmb.cam.ac.uk).

### Experimental Model and Subject Details
#### Animal strains
C57BL/6 wild type (WT) mice and *Rosa26*$^{LSL-tdTomato}$ transgenic mice (Jackson: Gt(ROSA)26Sor$^{tm14(CAG-tdTomato)}$) were used. All animal procedures were conducted in accordance with the UK Animals (Scientific procedures) Act 1986 and European Community Council Directive on Animal Care under project license PPL PCDD85C8A and approved by The Animal Welfare and Ethical Review Body (AWERB) committee of the MRC-LMB. Animals were housed in a 12 hours light/dark cycle with food and water ad libitum.

#### Cell lines
HEK293T cells were obtained from ATTC. HEK293T packaging cells expressing Rabies glycoprotein (HEK-GG) were generated by lentivirus infection with Lenti-$_{H2B}$GFP-2A- GlySAD and after 3 passages GFP expressing cells were selected by fluorescent activated cell sorting (FACS). HEK293T packaging cells expressing Rabies glycoprotein and TEV protease (HEK-TGG) were generated from HEK-GG by lentivirus infection with Lenti-puro-2A-TEV and selected, after 3 passages, with 1 µg/ml of puromycin added to the media for 1 week. HEK293T expressing TEV protease (HEK-TEVp) were generated by lentivirus infection with Lenti-puro-2A-TEV and selected, after 3 passages, with 1 µg/mL of puromycin added to the media for 1 week.

### Method Details
#### Design and generation of ΔG-Rabies and SiR plasmids
All Rabies and SiR plasmids were generated by Gibson cloning starting from pSAD-ΔG-F3 plasmid (*Osakada et al., 2011*) or SiR vectors we previously generated (*Ciabatti et al., 2017*), respectively. Engineered SiR vectors carrying d.C1349G or d.G1357T PEST-targeting mutations were produced by PCR amplification of the Rabies genome in 2 fragments starting from the end of N assembled using Gibson master mix (NEB).

The lentiviral vectors used to generate the packaging cells have been previously described (*Ciabatti et al., 2017*).

#### TEVp activity in packaging cells
Low passage HEK-TGG packaging cells were produced as previously described (*Ciabatti et al., 2017*). Briefly, HEK293T cells were infected with Lenti-GFP-2A-G and after three passages GFP expressing

cells were selected by fluorescent activated cell sorting (FACS). Cells were infected with Lenti-puro-2A-TEVp and amplified for two passages under 2 µg/ml of puromycin selection in 10% DMEM. This produced the HEK-TGG P0 line that was further amplified either in absence or presence of 1/2 µg/ml of puromycin selection for up to eight passages. Cells were split every 3 days at 1:6 dilution and every two passages TEVp activity was assessed by seeding 750 k cells in six-wells and transfecting a TEVp activity reporter (*Gray et al., 2010*) after 24 hr. Transfected cells were lysed in RIPA buffer after 24 hr and TEVp-dependent reporter cleavage was assessed by western blot staining for the V5 tag at the C-terminal of the TEVp activity reporter (monoclonal anti-V5 V8012, anti-mouse HRP-conjugated 32430). Western blots were imaged using a Chemidoc MP system (Bio-Rad) and the ratio of cleaved and uncleaved reporter was analysed using Image Lab software (Bio-Rad).

## Viral productions

SiR and ΔG-Rabies viruses were rescued from cDNA by the co-transfection of rabies genome vectors with pcDNA-T7, pcDNA-SADB19N, pcDNA-SADB19P, pcDNA-SADB19L, and pcDNA-SADB19G (*Osakada et al., 2011*) in HEK-TGG and HEK-GG cells, respectively, as previously described (*Ciabatti et al., 2017*).

For the recovery of high titer SiR and ΔG-Rabies, HEK-TGG or HEK-GG respectively were infected in 15 cm dishes at ~80% confluence with 3 ml of viral supernatant obtained as described in the viral screening section. Cells were split the day after infection and maintained for 1 or 2 days at 37 °C and 5% $CO_2$ checking daily the viral spreading when a fluorescent marker was present. Then, the media was replaced with 2% FBS DMEM and maintained for 2 days at 35 °C and 3% $CO_2$. Viral supernatant was collected, cell debris removed by centrifugation at 2500 rpm for 10 min followed by filtration with 0.45 µm filter and the virus concentrated by ultracentrifugation on a sucrose cushion as previously described (*Wickersham et al., 2007a*).

## Ontogenesis of revertant mutations during viral production

8 independent SiR viruses were rescued from cDNA as described in previous section. SiR RNA genomes were extracted from the infectious supernatants with RNeasy kit (Qiagen) following manufacturer's instructions and used to generate plasmid libraries for Sanger sequencing. To investigate the emergence of mutations during subsequent viral amplification rounds in vitro low passage HEK-TGG (HEKTGG P0), or high passage cells amplified in absence of puromycin pressure (HEK-TGG P8) were seeded in 10 cm dishes. At 60–70% confluence cells were infected with SiR supernatants obtained from cDNA at MOI=~2–3. The next day, cells were split at 1:2 dilution and maintained for 1 day at 37 °C and 5% $CO_2$ in 10% FBS DMEM. Then, media was replaced with 2% FBS DMEM and cells moved to incubation at 35 °C and 3% $CO_2$. Viral supernatants were collected after 2–3 days and used to infect fresh HEK-TGG P0 or HEK-TGG P8. The entire process was repeated for a total of 8 rounds of viral amplification. At each passage, 1 ml of supernatant was used to extract viral RNA genomes and generate libraries for NGS.

## Analysis of SiR accumulation of mutations during in vivo experiments

Sequence-verified revertant-free SiR virus was injected in CA1 region of the hippocampus of C57BL/6 wild type mice. After 1 week, mice were culled and the injected hippocampi manually dissected immediately. SiR genomes were obtained by homogenising the hippocampi with Tissuelyser II (Qiagen) and extracting the total RNA with RNeasy kit (Qiagen) according to manufacturer instructions. A total of 500 ng of RNA per hippocampus were reverse-transcribed using superscript IV kit (Invitrogen) and amplicons of N-TEVs-PEST were PCR-amplified to generate libraries for SMRT NGS sequencing.

## Sanger sequencing of SiR genomes

SiR genomic copies were extracted by concentrating 1 ml of infectious supernatant with Amicon Ultra-4 10 K filters in an Eppendorf 5810 R centrifuge at 4°C, 2500 *g* for 20' followed by RNeasy kit (Qiagen) extraction. RNA samples were treated with DNAse I (Invitrogen) for 15' at RT followed by inactivation at 65°C for 10'. Genomes were reverse-transcribed with SuperScript IV Reverse Transcriptase (Invitrogen) following manufacturer instructions using a primer complementary to the 5' leader sequence containing an 8 nt random barcode:

Leader_8barcode_: TCAGACGATGCGTCATGCNNNNNNNNNACGCTTAACAACCAGATC

cDNA samples were subjected to RNAse H treatment (NEB) followed by PCR amplification of a fragment corresponding to the entire coding sequence of N-TEVs-PEST and part of the P gene with Platinum SuperFi II Master Mix polymerase (denaturation for 30 s at 98°C; 25 cycles of amplification with 5 s at 98°C, 10 s at 60°C and 60 s at 72°C; 3 min at 72 for final extension) using primers:

Leader_PCR_Fw: ccaccgcggtggcggccgctcTCAGACGATGCGTCATGC
P_PCR_Rv: ctaaagggaacaaaagctgggtacCTTCTTGAGCTCTCGGCCAG

The obtained ~2 Kb amplicons were gel purified from 1% agarose gel using QIAquick Gel Extraction Kit (Qiagen) and cloned in pBluescript SK (+) (GenBank:X52325.1) digested KpnI – XbaI using Gibson assembly cloning method (NEB). 50 clones were purified and sequenced by Sanger method using M13_Fw and M13_Rv primers checking that each sequence carried a different 8 nt barcode.

## Single molecule real-time (SMRT) sequencing of SiR genomes

SiR supernatant preparations were first concentrated by centrifuging 1 ml of infectious supernatant in Amicon Ultra-4 10 K filters in an Eppendorf 5810 R centrifuge at 4 °C, 2500 *g* for 20', followed by RNA extraction using RNeasy kit (Qiagen). Purified viruses were directly extracted with RNeasy kit by adding 350 µl of RT lysis buffer to 5 µl of concentrated virus. RNA samples were treated with DNAse I (Invitrogen) for 15' at RT followed by inactivation at 65 °C for 10'. Genomes were retro-transcribed with SuperScript IV Reverse Transcriptase (Invitrogen) following manufacturer instructions using a primer complementary to the 5' leader sequence containing an adapter sequence and a 10 nt random barcode:

Pacbio_Leader_10barcode:CGAACATGTAGCTGACTCAGGTCACNNNNNNNNNNNCACGCTTAACAACCAGATC

cDNA samples were subjected to RNAse H treatment (NEB) followed by PCR amplification of a fragment corresponding to the entire coding sequence of N-TEVs-PEST and a fragment of the P gene with Platinum SuperFi II Master Mix polymerase (denaturation for 30 s at 98 °C; 25 cycles of amplification with 5 s at 98 °C, 10 s at 60 °C and 60 s at 72 °C; 3 min at 72 for final extension) using primers asymmetrically barcoded as shown below (list of the barcodes used for each sample can be found in *Tables 2 and 3*):

Pacbio_PCR_Fw: (16nt_barcode)CGAACATGTAGCTGACTCAGGTCAC
Pacbio_PCR_Rv: (16nt_barcode)AGTCGCCCCATATCCTCAGG

Barcodes:

bc1: TCAGACGATGCGTCAT
bc2: CTGCGTGCTCTACGAC
bc3: CATAGCGACTATCGTG
bc4: GCTCGACTGTGAGAGA
bc5: ACTCTCGCTCTGTAGA
bc6: TGCTCGCAGTATCACA
bc7: CAGTGAGAGCGCGATA
bc8: TCACACTCTAGAGCGA
bc9: GCAGACTCTCACACGC
bc10: GTGTGAGATATATATC
bc11: GACAGCATCTGCGCTC
bc12: CTGCGCAGTACGTGCA

The obtained ~2 Kb amplicons were gel purified from 1% agarose gel using QIAquick Gel Extraction Kit (Qiagen) followed by clean-up with QIAquick PCR purification kit (Qiagen). Purified barcoded amplicons from different viral preparations were combined in a single tube to obtain equimolar ratio and final concentration of ~50 ng/µl. SMRTbell libraries of pooled amplicons (up to 29 samples per library) were prepared using SMRTbell Template Prep Kit 1.0 (Pacbio) and Sequel chemistry v3 and sequenced on a PacBio Sequel SMRT cell with a 10 hr movie.

## Single-molecule real-time (SMRT) sequencing analysis

Pacbio Sequel II raw movies containing all subreads were used to generate high-fidelity circular consensus sequences (CCS) using pbccs program v4.2.0 (Pacific Biosciences,USA) (https://github.com/PacificBiosciences/ccs; *Pacific Biosciences, 2022*) with default settings (minimal number of passages 3, fidelity >98%). CCS reads were demultiplexed and assigned to each sample with the Lima program v1.11.0 (Pacific Biosciences,USA) (https://github.com/pacificbiosciences/barcoding/; *Pacific Bioscience, 2017*) using the asymmetric 16 nt barcodes added to the amplicons during PCR amplification (list of barcode combinations per sample in *Tables 2–3*). Duplicated sequences of the same genomic molecules were removed using the unique molecular identifiers (UMI) of 10 random nucleotides added during SiR genomes retrotranscription. Briefly, UMI tags were extracted from individual reads using UMI_tools v1.0.1 (https://github.com/CGATOxford/UMI-tools; *Smith et al., 2017*; *CGATOxford, 2023*) and used to generate families of reads from a single original genomic copy. For each family, the highest quality read was retained and the others discarded using dedup function of UMI_tools. Deduplicated reads were aligned to the reference using pbmm2 function v1.2.1 (Pacific Biosciences,USA) (https://github.com/PacificBiosciences/pbmm2/; *Pacific Biosciences, 2023*) and variants called using the ivar program v1.2.1 (https://github.com/andersen-lab/ivar; *Grubaugh et al., 2019*; *Andersen Laboratory, 2023*) using a minimum base quality of 20. Complete list of the identified mutations and number of reads above q>20 per base per sample can be found in *Tables 2 and 3*.

## TEVp-dependency of viral transcription

HEK and HEK-TEVp were seeded in glass bottom wells (μ-Slide 8 Well Glass Bottom, Ibidi) and infected when at ~70% confluence with SiR-nucGFP, SiR-S450X-nucGFP, SiRG453X-nucGFP or ΔG-Rabies-nucGFP. Live infected cells were imaged 48 hr post infection in an inverted confocal microscope (SP8 Leica) using a 10 x air objective with identical settings for all conditions to evaluate GFP expression levels.

## Immunohistochemistry

Mice were perfused with ice cold phosphate buffered saline (PBS) followed by 4% paraformaldehyde (PFA) in PBS. Brains were incubated in PFA overnight at 4 °C, rinsed twice with PBS followed by dehydration in 30% sucrose in PBS at 4 °C for 2 days. Then, brains were frozen in O.C.T. compound (VWR) and sliced at 35 μm on cryostat (Leica, Germany). Freefloating sections were rinsed in PBS and then incubated in blocking solution (1% bovine serum albumin and 0.3% Triton X-100 in PBS) containing primary antibodies for 24 hr at 4 °C. Sections were washed with PBS three times and incubated for 24 hr at 4 °C in blocking solution with secondary antibodies. Immuno-labelled sections were washed three times with PBS and mounted on glass slides. Antibodies used in this study were rabbit anti-RFP (Rockland, 600401–379, 1:2000) and donkey anti-rabbit Cy3 (Jackson ImmunoResearch, 711-165-152, 1:1000).

## Viral injections

All procedures using live animals were approved by the Home Office and the LMB Biosafety committee. For all experiments, adult mice >8 weeks were used. Mice were anesthetized with 3% isoflurane in 2 L/min of oxygen for the initial induction and then maintained with a flow of 1–2% isoflurane in 2 L/min of oxygen. Anesthetized animals were placed into a stereotaxic apparatus (David Kopf Instruments) and Rimadyl (2 mg/kg body weight) was administered subcutaneously (s.c.) as an anti-inflammatory. A small hole (500 μm diameter) was drilled and viruses were injected using a WPI Nanofil syringe (35 gauge) for injections in the hippocampus or a glass capillary for injections in the cerebral cortex. The syringe was left in the brain for 5 min before being retracted. SiR and Rabies viruses were injected at 3–6x10^8 infectious units/ml. For transsynaptic experiments, AAV-CMV-nucGFP-2A-TVA (AAV-TVA), AAV-hSyn1-TVAmCherry-2A-oG (AAV-TVA-G), AAV-hSyn1-TVAmCherry-2A-G(N2c) (AAV-TVA-G_N2c), AAV-hSyn1-nucFLAG-2a-TEVp (AAV-TEVp) were injected at ~3 × 10^12 genomic copies/ml. EnvA-pseudotyped SiR were injected at ~3 × 10^8 infectious units/ml for SAD-B19 strain and ~1–3 × 10^7 infectious units/ml for CVS-N2c strain. Up to a maximum volume of 500 nl of virus was injected in the following brain areas: hippocampus (AP: –2.45 mm, ML: 2 mm and DV: 1. 5 mm from bregma),

cerebral cortex (AP: –2.5 mm, ML: 2 mm and DV: 0,3 mm from brain surface), nucleus accumbens (AP: –1.3 mm, ML: 1.35 mm and DV: 4.7 mm from bregma).

## In vivo cytotoxicity analysis

SiR-CRE, SiR-G453X-CRE and ΔG-Rabies-CRE in vivo cytotoxicity was assessed by injecting 400 nl of purified viral preparations (at 3–6x10$^8$ infectious units/ml) in CA1 area of the hippocampus of $Rosa26^{LSL-tdTomato}$ mice. Animals were perfused at 1 week or 1–2 month p.i. and the brains were sectioned at the cryostat (35 μm). The entire hippocampus was sampled (by acquiring one slice every 4) by imaging infected neurons using a robot assisted Nikon HCA microscope mounting a 10 x (0.45NA) air objective and tdTomato positive hippocampal neurons counted using Nikon HCA analysis software. Cell survival was calculated by normalizing the total number of infected neurons to the 1 week time point.

## Transsynaptic spreading analysis

SiR transsynaptic spreading was assessed by injecting 500 nl of helper AAVs (at ~3 × 10$^{12}$ infectious units/ml) in the NAc of $Rosa26^{LSL-tdTomato}$ mice. After 3 weeks, animals were retargeted with 500 nl of purified EnvA-pseudotyped SiR-CRE, SiR-G453X-CRE or SiR-N2c-CRE. Animals were perfused at 1 month p.i. and the brains were sectioned at the cryostat (50 μm). The entire brain was sampled (by acquiring one slice every 4) by imaging infected neurons using a robot assisted Nikon HCA microscope mounting a 10 x (0.45NA) air objective and tdTomato$^+$ BLA neurons counted using Nikon HCA analysis software.

## Analysis of Rabies RNA in vivo

SiR-CRE genomic copies in vivo were evaluated over time by recovering the total RNA from SiR-injected hippocampi at different time points, as we previously described (*Ciabatti et al., 2017*). Briefly, the hippocampi were homogenized using a Tissuelyser II (QIAGEN) and processed accordingly to manufactory instruction with RNeasy kit (QIAGEN). A total of 500 ng of RNA per hippocampus were reverse-transcribed using superscript IV kit (Invitrogen) and analysed by quantitative PCR (Rotor-Gene Multiplex PCR) using probe assays against *Actb* and Rabies *N* gene. The Livak method was applied for quantification: the level of *N* at different time points was normalized to the expression of the *Actb* housekeeping gene ($\Delta CT = CT_{gene} - CT_{Actb}$) and the variation over time as fold change ($2^{-\Delta\Delta CT}$) to the 1 week time point ($\Delta\Delta CT = \Delta CT_{Time\ point} - \Delta CT_{1\ week}$).

## In vivo two-photon imaging

$Rosa26^{LSL-tdTomato}$ mice aged 3–4 months were injected with Dexafort at 2 μg/g, one day prior to surgery. Mice were anesthetized with Isofluorane (induction and maintenance at 3% and 2% in 3 L/min of oxygen, respectively) and injected subcutaneously with Vetergesic at 0.1 mg/kg. A metal head-post was affixed to the skull with Crown & Bridge Metabond. Epivicaine was splashed on the skull, and a 3 mm craniotomy was performed on the left hemisphere, centred at 2 mm lateral of the midline and 2.5 mm posterior of bregma. A total of 500 nl of virus with a titer of 4x10$^8$ was then delivered at the centre of the craniotomy, at a depth of 300 μm, and at a rate of 100 nl per minute using a manual hydraulic micromanipulator (Narishige). The craniotomy was finally sealed with a 3 mm round coverslip pressing on the brain, and affixed using Crown & Bridge Metabond. Mice were imaged weekly after surgery, under Isofluorane anaesthesia at 1.5% in 3 L/min of oxygen, with a two-photon microscope (Bergamo II, Thorlabs), equipped with a 16 x - 0.8 NA objective (Nikon). Infected cells were excited with a Ti:Sapphire pulsed laser at 1030 nm, with a power of around 20 mW (Mai TaiDeepSee, Spectra Physics). Emitted fluorescence was collected through a 607±35 nm filter (Brightline). For each mouse, a Z-stack was recorded, centred at the same anterior-posterior coordinate as the injection, but 1 mm closer to the midline in the lateral-medial axis. Imaging planes' pixel resolution was 2048x2048, and depth was sampled in steps of 1 μm. Z-stacks were 3d aligned across time points using a custom program written in Python, segmented into smaller fields of view, and filtered with a 3D mean filter of radius 2 pixels for x and y, and 5 pixels for z (Fiji). All cells at week 1 were labelled using FIJI, and their presence was manually assessed at later time points for the quantification of the survival rate.

## Quantification and statistical analysis

Mean values are accompanied by SEM. No statistical methods were used to predetermine sample sizes. In the hippocampal survival experiments animals were randomly assigned to each time point. Next generation sequencing datasets were analysed blindly. Otherwise, data collection and analysis were not performed blind to the conditions of the experiments. Statistical analysis was performed in Graphpad Prism and/or Matlab. Paired t-test and one-way ANOVA test were used to test for statistical significance when appropriate. Statistical parameters including the exact value of n, precision measures (mean ± SEM) and statistical significance are reported in the text and in the figure legends (see individual sections). The significance threshold was placed at $\alpha=0.05$.

# Acknowledgements

We thank Elena Williams for comments on the manuscript. We thank Jerome Boulanger for writing the script for the 3d-alignment of 2-photon recordings, Nicolas Alexandre for the help with the bioinformatic analysis of the NGS datasets, the Laboratory of Molecular Biology (LMB) workshops for the help with software and hardware development, and members of the Biological Service Group for their support with the in vivo work. This study was supported by the Medical Research Council (MC_UP_1201/2), the European Research Council (STG 677029 to MT), the European Union's Horizon 2020 research and innovation program with the Marie Sklodowska-Curie fellowship to DdM (894697), the Cambridge Philosophical Society and St. Edmund's College (University of Cambridge) with the Henslow Research Fellowship to AGR, the Rosetrees Trust with an MBPhD fellowship to HL (M598). For the purpose of open access, the MRC Laboratory of Molecular Biology has applied a CC BY public copyright licence to any Author Accepted Manuscript version arising. All data are stored on the LMB server. All materials described in this paper can be obtained upon reasonable request and for non-commercial purposes after signing a material transfer agreement (MTA) with the MRC.

# Additional information

### Competing interests

Ernesto Ciabatti: The SiR technology is patented by the UK Research and Innovation (WO2018203049A1). The other authors declare that no competing interests exist.

### Funding

| Funder | Grant reference number | Author |
|---|---|---|
| Medical Research Council | MRC-UP_1201/2 | Marco Tripodi |
| European Research Council | STG-677029 | Marco Tripodi |
| Marie Sklodowska-Curie Fellowship | Postdoctoral Fellowship 894697 | Daniel de Malmazet |
| Cambridge Philosophical Society and St. Edmund's College (University of Cambridge) | Henslow Research Fellowship | Ana González-Rueda |
| Rosetrees Trust | MBPhD fellowship M598 | Hassal Lee |

The funders had no role in study design, data collection and interpretation, or the decision to submit the work for publication.

### Author contributions

Ernesto Ciabatti, EC, conceived the project, designed the experiments, performed all the experiments and data analysis with the exception of the 2-photon recordings, wrote the manuscript, Conceptualization, Data curation, Formal analysis, Investigation, Methodology, Project administration, Resources, Supervision, Validation, Visualization, Writing – original draft, Writing – review and editing; Ana González-Rueda, Formal analysis, Investigation, Project administration, Resources, Validation,

Writing – review and editing, EC, conceived the project, designed the experiments, performed all the experiments and data analysis with the exception of the 2-photon recordings, wrote the manuscript; Daniel de Malmazet, Formal analysis, AG-R, Performed mouse surgeries, Assisted with preparing the manuscript, Investigation, Project administration, Resources, Validation, Writing – review and editing, Software; Hassal Lee, Formal analysis, Resources, Validation, Writing – review and editing, DDM, Performed mouse surgeries and the in vivo 2-photon imaging experiments and data analysis, Assisted with preparing the manuscript; Fabio Morgese, Data curation, Resources, Validation, Writing – review and editing, HL, performed molecular biology experiments, Assisted with preparing the manuscript; Marco Tripodi, Conceptualization, Methodology, FM, Provided technical support for cell culture and molecular biology, Assisted with reviewing and editing the manuscript, Visualization, Writing – review and editing

### Author ORCIDs
Ernesto Ciabatti https://orcid.org/0000-0001-9361-5992
Marco Tripodi http://orcid.org/0000-0002-6827-6690

### Ethics
This study was performed in strict accordance with the UK Animals (Scientific procedures) Act 1986 and European Community Council Directive on Animal Care. Animals were housed in a 12 hours light/dark cycle with food and water ad libitum.

### Decision letter and Author response
Decision letter https://doi.org/10.7554/eLife.83459.sa1
Author response https://doi.org/10.7554/eLife.83459.sa2

## Additional files

### Supplementary files
• MDAR checklist

### Data availability
Data generated during this study are included in the manuscript and supporting files. The viral vectors used in this study have been previously described (*Ciabatti et al., 2017*) and are available from Addgene. The raw NGS datasets have been deposited into NCBI's Sequence Read Archive (SRA) and are accessible through accession number PRJNA888353.

The following dataset was generated:

| Author(s) | Year | Dataset title | Dataset URL | Database and Identifier |
|---|---|---|---|---|
| Ciabatti E | 2022 | SiR genomic stability | https://www.ncbi.nlm.nih.gov/sra/?term=PRJNA888353 | NCBI Sequence Read Archive, PRJNA888353 |

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

# Appendix 1

## Appendix 1—key resources table

| Reagent type (species) or resource | Designation | Source or reference | Identifiers | Additional information |
|---|---|---|---|---|
| Strain, strain background (mouse *Rosa26*<sup>LSL-tdTomato</sup>) | B6.Cg*Gt(ROSA)26Sor*<sup>tm14(CAG-tdTomato)Hze</sup>/J | Jackson Labs (H.Zeng) | 007914 | |
| Cell line (*Homo-sapiens*) | HEK293T | ATTC | CRL-3216 | |
| Cell line (*Homo-sapiens*) | HEK-GG | This paper | | See Methods. |
| Cell line (*Homo-sapiens*) | HEK-TGG | This paper | | See Methods. |
| Cell line (*Homo-sapiens*) | HEK-TEVp | This paper | | See Methods. |
| Recombinant DNA reagent (plasmid) | pLenti-puro-2A-TEV | *Ciabatti et al., 2017* | Addgene: 99610 | |
| Recombinant DNA reagent (plasmid) | pLenti-$_{H2B}$GFP-2A-GlySAD | This paper | | See Methods. |
| Recombinant DNA reagent (plasmid) | pSiR-CRE | This Paper | | Derived from Addgene: 99608. See Methods. |
| Recombinant DNA reagent (plasmid) | pSiR-S450X-nucGFP | This Paper | | Derived from Addgene: 99608. See Methods. |
| Recombinant DNA reagent (plasmid) | pSiR-G453X-nucGFP | This Paper | | Derived from Addgene: 99608. See Methods. |
| Recombinant DNA reagent (plasmid) | pSiR-G453X-CRE | This Paper | | Derived from Addgene: 99608. See Methods. |
| Recombinant DNA reagent (plasmid) | pSiR-N2c-CRE | *Lee et al., 2023* | Addgene: 194456 | |
| Recombinant DNA reagent (plasmid) | pΔG-Rabies-CRE | This paper | | See Methods. |
| Recombinant DNA reagent (plasmid) | pAAV-CMV-nucGFP-2A-TVA (AAV-TVA) | This paper | | See Methods. |
| Recombinant DNA reagent (plasmid) | pAAV-hSyn1-TVAmCherry-2A-G(N2c) (AAV-TVA-G_N2c) | *Lee et al., 2023* | Addgene: 194354 | |
| Recombinant DNA reagent (plasmid) | pAAV-hSyn1-TVAmCherry-2A-oG(AAV-TVA-G) | This paper | | Derived from Addgene: 194354. See Methods. |
| Antibody | Anti-V5 tag antibody (mouse monoclonal) | Sigma Aldrich | V8012 | 1:5000 dilution |
| Antibody | anti-Mouse IgG (H+L) HRP-conjugated (goat polyclonal) | Invitrogen | 32430 | 1:2000 dilution |
| Sequence-based reagent | qPCR assay against *Actb* gene (HEX-conjugated) | IDT | Mm.PT.39a.22214843.g | |
| Sequence-based reagent | qPCR assay against Rabies N gene (6-FAM-conjugated) | IDT | | <u>FW</u>: CAGGTTCTCTGGTGGAGATAAA<br><u>Probe</u>: TGACAGGAGGCATGGAACTGACAA<br><u>RV</u>: CTCAAGAGAAGACCGACTAAGG |

