## [Editor Report]

The authors previously described a viral tool termed 'self-inactivating rabies' to trace neural circuits with minimized cell toxicity. However, this tool acquired mutations during passage which revert the tool to previous toxicity levels. This manuscript provides clarification on how to propagate and use the tool to minimize toxicity-promoting mutations.

---

## [Decision Letter]

**Decision letter after peer review:**

Thank you for submitting your article "Genomic stability of Self-inactivating Rabies" for consideration by *eLife*. Your article has been reviewed by 3 peer reviewers, and the evaluation has been overseen by a Reviewing Editor and Catherine Dulac as the Senior Editor. The following individuals involved in the review of your submission have agreed to reveal their identity: Ed Callaway (Reviewer #1); Chun Xu (Reviewer #2); Keisuke Yonehara (Reviewer #3).

Essential revisions:

1) It is absolutely essential that the authors demonstrate the unmutated virus retains the ability to serve as an effective trans-synaptic neuronal tracer. In particular, it is critical to address the comments of reviewer 1 in full.

*Reviewer #1 (Recommendations for the authors):*

Please add descriptions of any attempts at transsynaptic tracing with non-mutated SiR. Also, after adding negative results (if that is the result) the discussion could consider how the use of TEV expression might allow transsynaptic tracing.

*Reviewer #2 (Recommendations for the authors):*

It feels like that it would be difficult to find a good balance between rabies production with high titer and passage number of cell lines. It would be great to see discussion and potential analytic estimation on how to produce a good titer of rabies vectors with negligible amount of mutated rabies in vivo.

*Reviewer #3 (Recommendations for the authors):*

1. The stars (*) in the legend for Figure 1B is a bit complicated because there are two types, marked in red color and black color. It is better to describe it more clearly.

2. What does HEK-TEVp mean in Figure S3? Please describe this cell line in the method section.

3. In the method section, the description of Western blot analysis looks insufficient. The authors need to add more details with the identifier of primary/secondary antibodies.

---

## [Author Response]

Essential revisions:Reviewer #1 (Recommendations for the authors):Please add descriptions of any attempts at transsynaptic tracing with non-mutated SiR. Also, after adding negative results (if that is the result) the discussion could consider how the use of TEV expression might allow transsynaptic tracing.

As discussed above we have now given a detailed report of the spreading efficiency of the non-mutated SiR comparing it to the spreading following the addition of TEVp to the starting cells. Furthermore, we also compared these two conditions (+ and – TEVp) for the newly generated Sir2.0 based on the CVS-N2c strain.

Reviewer #2 (Recommendations for the authors):It feels like that it would be difficult to find a good balance between rabies production with high titer and passage number of cell lines. It would be great to see discussion and potential analytic estimation on how to produce a good titer of rabies vectors with negligible amount of mutated rabies in vivo.

We understand the importance of defining a reliable protocol to generate unmutated SiR. For this reason, we have clarified that our starting P0 conditions for packaging cells and stock SiR viruses (Figure 2C and Figure 2E, respectively) were equivalent to already amplified stocks ready for viral production. Since viral production following our original protocol would only add 1-2 passages, SiR viruses will fall well below the identified threshold point for accumulation of detectable revertant mutations. We have showed in this work that SiR viruses produced accordingly do not show revertant mutations in vitro, after injection in vivo and can spread transsynaptically.

Reviewer #3 (Recommendations for the authors):1. The stars (*) in the legend for Figure 1B is a bit complicated because there are two types, marked in red color and black color. It is better to describe it more clearly.

Thanks, we have now updated this changing one type of star to another symbol.

2. What does HEK-TEVp mean in Figure S3? Please describe this cell line in the method section.

We thank the reviewer for highlighting the lack of clarity of this point. We have now updated the method section and the Key Report Table to better describe the nature of the cell line.

Changes in the manuscript: The updated paragraph is provided here:

Methods: Cell lines

“HEK293T expressing TEV protease (HEK-TEVp) were generated by lentivirus infection with Lenti-puro-2A-TEV and selected, after 3 passages, with 1 µg/mL of puromycin added to the media for 1 week.”

3. In the method section, the description of Western blot analysis looks insufficient. The authors need to add more details with the identifier of primary/secondary antibodies.

Once again, thanks for the suggestion. We agree that the previous version of the manuscript lacked in clarity on this point and we have now updated the method section and the Key Report Table.

Changes in the manuscript: The updated paragraph is provided here:

Methods: TEVp activity in packaging cells

“Transfected cells were lysed in RIPA buffer after 24 hrs and TEVp-dependent reporter cleavage was assessed by western blot staining for the V5 tag at the C-terminal of the TEVp activity reporter (monoclonal anti-V5 V8012, anti-mouse HRP-conjugated 32430). Western blots were imaged using a Chemidoc MP system (Biorad) and the ratio of cleaved and uncleaved reporter was analysed using Image Lab software (Biorad).”